

# Accelerating covering array generation by combinatorial join for industry scale software testing

Hiroshi Ukai[1,2], Xiao Qu[3], Hironori Washizaki[1] and Yoshiaki Fukazawa[1]

[1] Waseda University, Tokyo, Japan
[2] Rakuten Group, Inc., Tokyo, Japan
[3] Independent Researcher, Raleigh, NC, United States of America

## ABSTRACT

Combinatorial interaction testing, which is a technique to verify a system with numerous input parameters, employs a mathematical object called a covering array as a test input. This technique generates a limited number of test cases while guaranteeing a given combinatorial coverage. Although this area has been studied extensively, handling constraints among input parameters remains a major challenge, which may significantly increase the cost to generate covering arrays. In this work, we propose a mathematical operation, called "weaken-product based combinatorial join", which constructs a new covering array from two existing covering arrays. The operation reuses existing covering arrays to save computational resource by increasing parallelism during generation without losing combinatorial coverage of the original arrays. Our proposed method significantly reduce the covering array generation time by 13–96% depending on use case scenarios.

## INTRODUCTION

Modern software systems consist of multiple components, each of which is composed of several elements, where each element has multiple parameters. Due to the combinatorial explosion, exhaustively testing all possible combinations of inputs is impractical during product testing even if all possible values for each parameter are limited by equivalence partitioning. One way to handle this situation is to employ a technique called Combinatorial Interaction Testing (CIT) (*Kuhn, Kacker & Lei, 2013*). CIT applies a mathematical object called a covering array to incorporate all possible $t$-way combinations of parameter values as a test input to a certain system under test (SUT). The variable $t$, which is called testing strength (we will use "strength" for short hereafter), guarantees all the possible value combinations of $t$ parameters to be covered in the test. Previous studies intensively investigated how to reduce both the size of a covering array size and its generation time.

Applying CIT techniques to the real-world software products remains a challenge. First, real-world software products have numerous input parameters, resulting in a very long time

Corresponding author
Hiroshi Ukai,
hiroshi.ukai@akane.waseda.jp

to generate a covering array of very large size. Second, a value for each parameter cannot be assigned independently. Values must be chosen to satisfy a certain set of conditions, which are called *constraints*. Handling constraints can make the size and generation time of a covering array impractical. At the same time, constraints to describe a software product's specification may become complicated, further increasing the size and time even more.

To mitigate this situation, it is more efficient to apply a "divide-and-conquer" approach instead of generating it to generate, a covering array for a software product with numerous which has numerous parameters under complex constraints. This approach splits a set of parameters into multiple groups, generates covering arrays for each group, and combine them into one. It requires constructing a new covering array from existing ones.

Methods to construct a new covering array from existing ones are relatively less studied (*Kampel, Garn & Simos, 2017*; *Kruse, 2016*; *Zamansky et al., 2017*; *Ukai et al., 2019*). Theey can be divided into three categories. The first category constructs a combined array from the input arrays by viewing each input array as a parameter whose values are its rows (*Kampel, Garn & Simos, 2017*). The second category reuses and *extends* an existing covering array (*Cohen et al., 1997a*; *Czerwonka, 2006*; *Nie & Leung, 2011*). Many popular tools (*Kuhn, Kacker & Lei, 2008*; *Cohen et al., 1997a*; *Czerwonka, 2006*) have been implemented in this category. These tools can handle new parameters that are not present in the initial covering array and generate an output that covers all combinations. This feature is usually called 'seeding' or 'incremental generation'. The third category applies is an operation called *combinatorial join* (*Ukai et al., 2019*), which generates a new covering array by combining rows in input covering arrays while ensuring all value combinations across input arrays are covered.

By separating the implementation method from the operation introduced in the third method, in this paper we present a design of a novel algorithm to implement the *combinatorial join* operation, which is called "weaken-product based combinatorial join". Additionally, we evaluate the efficiency and practicality of our method by comparing it to the conventional methods (*i.e.*, new generation and incremental generation) implemented in a popular tool called ACTS (*Kuhn, Kacker & Lei, 2008*). Our experiments measure the generation time for modeled systems with various constraints and sizes. Since our approach constructs a new covering array from existing ones without creating a new row, it has minimal opportunities to reduce the size of output. Thus, we also conduct experiments to ensure that the the increased output size ("size penalty") remains reasonable. Our approach significantly reduces the generation time by 33–88% for strength of 2 or 3 while the size penalty remains practical.

Our approach delivers other benefits. First, in a real software project, it is not practical to conduct combinatorial testing in the same strength regardless of each component's importance. A variable strength covering array (VSCA) is a mathematical object to handle this situation (*Cohen et al., 2003*; *Cohen et al., 1997b*), where subsets of attributes in the entire array may have higher strength than the others. Various methods to construct it are proposed (*Bansal et al., 2015*; *Wang & He, 2013*). Since our combinatorial join operation is transparent to the input covering array's strength, if we give covering arrays of strength

$u$ as the input and perform the operation in strength $t$, it will result in a VSCA. The results of our study (RQ4) show a 10%–60% reduction in generation time.

Second, in some other practical situations, it is possible and desired to reuse test oracles designed for an earlier testing phase in a later one (*Ukai et al., 2019*). However, existing CIT tools can only reuse test oracles defined for only one single component among all. This is achieved by a technique called "incremental generation", which is the second category of the methods to construct a new array from existing arrays (*Kuhn, Kacker & Lei, 2013*). The test oracle reuse is very limited in this method because the incremental generation allows to use only one covering array as the seeds and therefore a completely new covering array is generated for attributes that are not included in the seeds. This forces testers to redefine new test oracles for those attributes not in the seeds even if they already have ones for a covering array generated from the attributes outside the incremental generation procedure. The combinatorial join operation allows to give two input covering arrays as the inputs without creating any new row from scratch and it will enhance possibility to reuse test oracles for testers. In this work (RQ3), we define the operation using the characteristics of its inputs and outputs declaratively so that one can provide other implementation of the operation by satisfying the definitions. We also qualitatively discuss the conditions and assumptions, where test oracle reuse by the combinatorial join is able to deliver benefits for testers.

Furthermore, in order to describe a software product's specification, sometimes a sufficiently high-level abstraction of constraints is required and otherwise the constraint definition will become impractically complicated. Such a capability is provided only by limited tools. Various tools, which generate covering arrays of a specified strength under constraints, have been developed and proposed, such as ACTS (*Kuhn, Kacker & Lei, 2008*), PICT (*Czerwonka, 2006*), JCUnit (*Ukai 2007*), *etc.*, each of which has its own strengths and weaknesses. Among all of them, ACTS is utilized most widely because of its rich functionality and outstanding performance in both time and the size of its output, on the other hand, its capability to model constraints only provides the most basic operators and data types. Nevertheless, to the best of our knowledge, no single tool is capable of handling all of these challenges mentioned above in a large scale software product development. With the combinatorial join operation, we can consider an approach where parameters are split into groups and the final covering array is constructed by combining sub-covering arrays each of which is generated by an optimal tool for each group. In this work (RQ3, RQ4), we examine whether this approach is beneficial and possible in what circumstances, qualitatively.

In summary, the contributions of this work are as follows, which altogether enhance the applicability of CIT toward the larger and more complex software products in the real world.

- Our proposed algorithm and implementation of combinatorial join makes CIT technique more efficient and flexible in large scale software systems with complex constraints.

   – we improved our previous work by introducing a new algorithm, where the strengths of the input covering arrays are reduced and then connected so that the desired strength in the output is achieved.
   – our tool generates covering arrays (with same strength) and VSCAs with constraints faster than a very popular tool (RQ1).

- We have evaluated how the size of generated test suite behaves under various conditions (RQ2).
- Our tool makes it possible to reuse test oracles without extra manual effort (RQ3).
- Our tool makes it possible to use multiple tools to generate one test suite, by taking advantage of each tool to generate of sub-arrays in different situations (RQ4).

The remainder of this paper is organized as follows. In 'Background and Related Works', we introduce the background and related work of CIT technique and its related topics, such as constraint handling support, incremental generation, and variable strength covering arrays. In 'Weaken-product-based Combinatorial Join Technique', we describe our algorithm to implement the combinatorial join operation and provide proofs that it can generate a new covering array from two given covering arrays. Then, we conduct experiments to acquire the performance characteristics of an existing tool and examine whether our approach is beneficial. In 'Evaluation, Results', we evaluate different use cases, parameter sizes, and constraint sets to determine whether our method accelerates covering array generation and realizes practical covering array sizes. We finish in 'Conclusion', by discussing the efficiency and benefit of our approach with its limitations and future works.

# BACKGROUND AND RELATED WORKS

## Combinatorial interaction testing

Combinatorial Interaction Testing (CIT) technique generates a test suite that contains all the possible combinations of values among any $t$ parameters for a system under test. A test suite generated by a CIT tool is called a *coveringarray*. It is denoted as $CA(N; t, k, v)$, where $N$ is the number of rows, $t$ is called testing *strength*, $k$ is the number of columns (*i.e.*, parameters), and $v$ is the number of possible values for each parameter.(here we assume each parameter has the same number of possible values) $k$ and $v$ are called *degree* and *order* respectively (*Kuhn, Kacker & Lei, 2013*).

CIT is useful to shrink the full Cartesian product space of a set of parameters, which becomes impractical for large-scale applications, into a reasonable test suite. The test suite generated by a CIT tool is called a *covering array*.

The most common type of covering array in CIT is pairwise ($t = 2$) in which all two-way combinations of parameter values are tested together in at least one test case. Numerous algorithms have been proposed to generate such artifacts (*Nie & Leung, 2011*; *Anand et al., 2013*), from greedy algorithm (*e.g.*, AETG (*Cohen et al., 1997a*), IPOG (*Lei et al., 2008*, and PICT (*Czerwonka, 2006*)), simulated-annealing (*Garvin, Cohen & Dwyer, 2011*) to heuristic search-based technique (*Shiba, Tsuchiya & Kikuno, 2004*).

CIT has been applied to various applications including GUI testing, configuration-aware system testing (such as product line testing), and unit testing. A study in 2018

reported 40 commercial or open source tools have been developed to generate CIT test suites (*Czerwonka, 2018*).

The generation of a covering array has been extensively studied, to minimize the size of a covering array, to deal with constraints defined in a test model(*Grindal, Offutt & Mellin, 2006*; *Wu et al., 2019*), or to generate a covering array by extending an existing covering array (*i.e.* incremental generation, *Kampel, Garn & Simos (2017)*; *Kruse (2016)*; *Zamansky et al. (2017)*; *Ukai et al. (2019)*), rather than from scratch.

## Constraint support by existing tools

In a practical software system each parameter cannot be assigned independently. Instead, parameter values must be selected so that a certain set of conditions are satisfied. Such conditions are called *constraints*. For example, when we test a system equipped with web-based GUI, **OS** (Windows, Mac OS, Linux, *etc.*) and **browser** (Edge, Safari, Chrome, Firefox, *etc.*), **OS** and **browser** are parameters and their values specified in the parentheses are different settings that a user may access to the system. In a test case where Safari or Edge is chosen as the parameter **browser**, Linux cannot be assigned as an **OS** parameter. This is an example of a constraint. If a test case violating a constraint is introduced in a test suite, it will not cover the expected combinations of values, even those not related to the constraint, because this whole test case will not be valid. As a result, the combinatorial coverage of the whole test suite will be damaged. Specifically in our example, when we create a test case where Safari is chosen for **Browser** and Linux for **OS**, the test case is expected to cover valid value combinations for other parameters such as **Font**, **Language**, **Timezone**. Now the test case is violating a constraint about **OS** and **browser** and it makes the entire test case invalid. This means combinations for the other parameters (**Font**, **Language**, *etc.*) will not be executed unless they are accidentally covered by other test cases. Accidental coverage occurs much less frequently than one may expect because the CIT minimizes the number of tests cases to avoid repeating the same value combinations. Constraints are often denoted in a format of tuples that are forbidden to be present in the output covering array. For example, the constraint that *Linux* of *OS* cannot be tested together with *Safari* of browser is denoted as $(OS_{Linux}, browser_{Safari})$, where *OS* and *browser* are names of parameters and *Linux* and *Safari* are their values.

ACTS has a superior performance with respect to both generation speed of covering arrays and covering arrays size without constraints, based on a comparison between various tools conducted by *Kuhn, Kacker & Lei (2013)*. For example, when ACTS generates a covering array of $CA(2, 2, 100)$ with no constraint, it takes less than 1.0 [sec] and the size of the generated covering array is 14. Another popular tool, PICT can generate a covering array of $CA(2, 2, 100)$ in less than 1.0 [sec] with 15 rows, but it shows quite unpractical performance when a complex constraint set is present (*Czerwonka, 2016*).

However, in terms of ability to define or describe complicated constraints and parameters (we call it *flexibility*), other tools (*e.g.*, PICT and JCUnit) outperform ACTS. Flexibility of defining constraints is less researched than performance of generating covering arrays under constraints, but it is very important in practice. The effort to define constraints is necessary to model relationships between parameters and such a model sometimes becomes

so complex that it requires a notation as powerful as a popular programming language, where products under testing are developed. On the other hand, introducing such a rich feature into the notation to describe constraints makes it difficult to implement an efficient covering array generator because constraint handling sometimes relies on an external SAT solver, which is not as powerful as a general purpose programming language such as Java.

In short, no single CIT tool provides superior performances for all requirements such as size, speed, and flexibility in constraint handling, simultaneously.

We next describe three tools studied in our research, ACTS, PICT, and JCUnit, with a focus on their different characteristics in defining constraints.

### ACTS

ACTS supports four data types, which are bool, number, enum, and range. The following code block contains examples to define factors of those types.

```xml
<Parameters>
  
    <values>
      <value>elem1</value>
      <value>elem2</value>
    </values>
    <basechoices />
    <invalidValues />
  
  
    <values>
      <value>0</value>
      <value>100</value>
      ...
      <value>2000000000</value>
    </values>
  
  
    <values>
      <value>true</value>
      <value>false</value>
    </values>
  
  
    <values>
      <value>0</value>
      <value>1</value>
      <value>2</value>
      <value>3</value>
    </values>
  
  ...
</Parameters>
```

ACTS has a very primitive set of mathematical and logical operators that can be used in constraint definitions. For instance, it supports $<$ but not $>$. Although $>$ can be expressed using the $<$ and negate (!) operators, it complicates the readability of the constraint definition. Also it lacks conditional operators such as a ternary operator or if-then-else structure. This can also be substituted with a combination of supported logical operators such as negate and conjunction or negate and disjunction, however, such substitutions also complicate the readability.

In our experience, lacks of those operators result in impractical constraint definitions that are hard to read and understand. Following is an example to define a constraint with ACTS.

```
<Constraints>
    <Constraint text="l01 <= l02 || l03 <= l04
                || l05 <= l06 || l07<= l08 || l09 <= l02">
      <Parameters>
        
        
        
        
        
        
        
        
        
      </Parameters>
    </Constraint>
  </Constraints>
```

This is equivalent to the following formula:

$$l01 <= l02 || l03 <= l04 || l05 <= l06 || l07 <= l08 || l09 <= l02 \tag{1}$$

We can also define a constraint that checks if values satisfy a certain formula using mathematical operators such as $+$, $-$, $*$, and $/$.

### PICT

PICT supports a couple of data types, which are enum and numeric. Following is an example to define a test model in PICT (*Czerwonka, 2015*).

```
PLATFORM: x86, ia64, amd64
CPUS:      Single, Dual, Quad
RAM:       128MB, 1GB, 4GB, 64GB
HDD:       SCSI, IDE
OS:        NT4, Win2K, WinXP, Win2K3
IE:        4.0, 5.0, 5.5, 6.0
```

Unlike ACTS, PICT does not support data types such as bool or range, but this is not an essential drawback of the tool, because these types can be represented by enum with appropriate symbols as an alternative, and such substitutions will not affect readability severely. For constraint handling, PICT provides quite readable notation as shown below.

```
IF [PLATFORM] in {"ia64", "amd64"} THEN [OS] in {"WinXP", "Win2K3"};
IF [PLATFORM] = "x86" THEN [RAM] <> "64GB";
```

In this example, PICT uses IF-THEN-ELSE structure to define constraints. Without this structure, the same constraints need to be converted in a more complicated way, as shown below. This is how constraints are defined using ACTS. Though such conversion is not difficult, it is usually an error prone manual process. Moreover, as we pointed out already, the converted constraints are hard to read and understand by engineers, since they lost their original designs mapped back to the system test model.

```
! PLATFORM = ia64 && ! PLATFORM = amd64 || (OS = WinXP || Win2K3)
! PLATFORM = x86 || ! RAM = 64GB
```

On the other hand, however, PICT does not support mathematical operators between parameters, hence it cannot define a constraint that requires such operators, which can be done by ACTS.

### JCUNIT

Given that both ACTS and PICT have their own limitations in constraint definition, we introduced a new tool in our previous work *Ukai (2007)*.

JCUnit allows a user to define a constraint as a method written in Java, which takes values for factors as parameters and returns a boolean value. The following example defines

a constraint for a set of integer parameters a, b, and c. These parameters are coefficients in a quadratic equation, $ax^2 + bx + c$, and the constraint checks if this equation has a solution in real.

```
@Condition ( constraint  =  true )
public  boolean  discriminantIsNonNegative (
    @From ( "a" )  int  a ,
    @From ( "b" )  int  b ,
    @From ( "c" )  int  c )  {
  return  b  *  b  −  4  *  c  *  a  >=  0;
}
```

For programmers, this style delivers a benefit that they can define constraints in the same way as they write their product code, and the definition can be as readable as a regular Java language program. However the tool is unable to employ external tools such as SAT libraries because the constraints are expressed as a normal Java program that external tools do not understand. Hence, it needs to rely on its internal logic to handle constraints. This makes overall constraint handling cost less efficient, although it is still faster than PICT (*Ukai, 2017*). JCUnit also allows any values as levels for a factor as long as they are an appropriately implemented Java object.

```
@ParameterSource
public  Simple . Factory < Integer >  depositAmount ()  {
  return  Simple . Factory . of ( asList (100 ,  200 ,  300 ,  400 ,  500 ,  600 ,  −1));
}

@ParameterSource
public  Regex . Factory < String >  scenario ()  {
  return  Regex . Factory . of ( "open  deposit ( deposit | withdraw | transfer ){0 ,2} getBalance ");
}
```

The code block shown above illustrates how a normal factor (*e.g.*, depositAmount) and a regex type factor (*e.g.*, scenario) can be defined. ''depositAmount'' is a factor of an Integer type defined in a method with the same name, which has 100, 200, 300, 400, 500, 600, and −1 as its levels. As mentioned already any Java object can be used as a possible value (level) of a parameter (factor), users are able to use methods defined for the class in the constraint definition. This makes it possible to define a constraint which examines whether the length of a string parameter exceeds a certain amount or not, for instance, and contributes to the readability of the constraint definition.

In addition, it provides a special data type ''regex'', which produces a set of factors that represents a sequence of values conforming to a given expression (''scenario'' method in the example). Through this method, a user can access a parameter ''scenario'' whose possible values are list of Strings, which are [open, deposit, getBalance], [open, deposit, deposit, getBalance], [open, deposit, withdraw, getBalance], *etc.* This feature is implemented by expanding the parameter into multiple small factors, each of which represents an element in the list and constraints over them. JCUnit internally generates those factors and constraints and constructs a covering array from them.

## Reuse covering arrays

Generating a covering array is an expensive task, especially when executed under complex constraints, a higher strength than two, and/or there are a number of parameters. Since a large software system can have a complex internal structure and hundreds or even more parameters, divide-and-conquer approach is desirable. If the time of covering array generation grows non-linearly along with the number of parameters $n$ (*e.g.*, $n^2$, $n^3$),

this approach may accelerate the overall generation because a set of parameters can be divided into multiple groups. Dividing into groups can prevent an explosive increase in the generation time for each group, even if there is overhead to recombine them into one .

To enable such an approach, a method to construct a new covering array reusing existing ones is necessary. However, such methods are not as well studied as methods to generate covering array from scratch (*Kampel, Garn & Simos, 2017*; *Kruse, 2016*; *Zamansky et al., 2017*; *Ukai et al., 2019*).

The most popular method for reusing a covering array is a feature called "seeding" (*Cohen et al., 1997b*). Seeding takes an existing covering array and parameters to be added as inputs. Hereafter, we refer to this method as *incremental generation*. This allows mandatory combinations to be specified for a tool, minimizing changes in the output. Minimizing changes is important because the output, which represents a test suite, sometimes contains fundamental parameters that are expensive to control such as OS or filesystem to be used in test execution. Popular tools for CIT such as ACTS (*Kuhn, Kacker & Lei, 2008*), PICT (*Czerwonka, 2006*), and JCUnit (*Ukai 2007*) can add parameters not presented in an initial covering array and generate an output as by assigning values to them so that the combinations between the values of the given parameters and the existing ones are covered. However, this limits reuse of only one covering array.

Another approach is to apply a CIT technique by setting each input covering array is a parameter whose rows are possible values (*Zamansky et al., 2017*). One drawback to this approach is that it makes the final array's size larger than $M \times N$, where $M$ is the maximum array's size in the input and $N$ is the second maximum's size. This results in an output with an impractical size for large-scale software product development.

As a third approach, in our previous work, we proposed an operation called *combinatorial join* (*Ukai et al., 2019*) to reuse covering arrays. Combinatorial join assumes that input arrays are already covering arrays and a new row in the output is created by connecting rows in the input arrays so that the entire output becomes a new covering array which has all the parameters to test. *Ukai et al. (2019)* presented an implementation of the combintorial join operation based on a covering array generation algorithm called IPOG (*Lei et al., 2008*). However, the implementation was impractically expensive in terms of time and memory usage when there are more than 100 parameters or strength $t$ exceeds 2.

## Variable strength covering array

A variable strength covering array (VSCA) is a covering array where the strength $t$ can be different depending a set of parameters among all of them (*Cohen et al., 2003*). It is considered useful to apply VSCA for testing a system which consists of multiple components since some components are more critical than others in a large system. Methods to generate VSCA have been proposed in related work (*Bansal et al., 2015*; *Wang & He, 2013*).

As introduced later in 'Weaken-product-based Combinatorial Join Technique', our proposed combinatorial join operation can also generate a VSCA, because this approach guarantees to include all the rows in input arrays at least once, if one array has a higher strength than the other, the portion corresponding to the array will have the same strength as the input.

# WEAKEN-PRODUCT-BASED COMBINATORIAL JOIN TECHNIQUE

A real-world software product has numerous parameters, which causes a combinatorial explosion when conducting a fully exhaustive testing. A CIT technique provides a way to handle this situation while guaranteeing reasonable coverage over all combinations of possible parameter values. However, generating a test suite employing the CIT technique is an expensive process, particularly when complicated constraints over the parameters are present. One approach to solve this issue is to generate test suites for components in the system separately and then combine them into one. The *combinatorial join* operation can realize this idea as it takes two inputs *LHS* (Left Hand Side) and *RHS* (Right Hand Side) and generates one output covering array from them. *LHS* and *RHS* are pre-generated covering arrays and there is no constraint across them as the precondition of the operation.

This output array contains all the rows from *LHS* and *RHS*, covers all the t-way combinations across them, but not include any extraneous rows that are not found in *LHS* or *RHS*. In a simple case, the input covering arrays (*i.e.*, *LHS* and *RHS*) can be test suites generated for individual components. But when we employ the technique to apply "divide-and-conquer" approach with this technique for a large scale software product, we can split the parameters of the product into two groups as *LHS* and *RHS*, regardless of actual components. The split needs to be done in a way that parameters from *LHS* and *RHS* may not exist together in one constraint. It is also preferable to make both *LHS* and *RHS* have the same number of parameters and constraints in order to maximize the benefit of parallelism.

The technique weaken-product based combinatorial join proposed in this paper implements the operation, which has practical performance for industry scale software developments.

The method proposed in our previous work (*Ukai et al., 2019*) intended to achieve the same goal of this work, but it was based on an algorithm similar to IPO and worked only when strength = 2 and degree is less than hundred in practice. The method proposed in this paper improves the previous work in several ways: (1) it constructs a new covering array from input arrays so that the strengths of the input arrays can be reduced, hence the cost of generating the input arrays are reduced. (2) the new method is studied for strength greater than 2 and it handles degrees as large as one thousand.

This approach will be beneficial for systems like listed below:

- A system consists of multiple components whose parameters are too expensive to change for each test case, generating a covering array from existing ones provides an efficient way of testing while guaranteeing combinatorial coverage over the entire system.
- A peer-to-peer communication system is tested and we desire to detect failures triggered by combinations of such parameter values across computers, for instance, OSes, browsers, languages, regions, and time-zones.

As mentioned earlier, constraint handling is supported by various tools but in different ways, where each tool has its own strengths and weaknesses. Since the combinatorial join

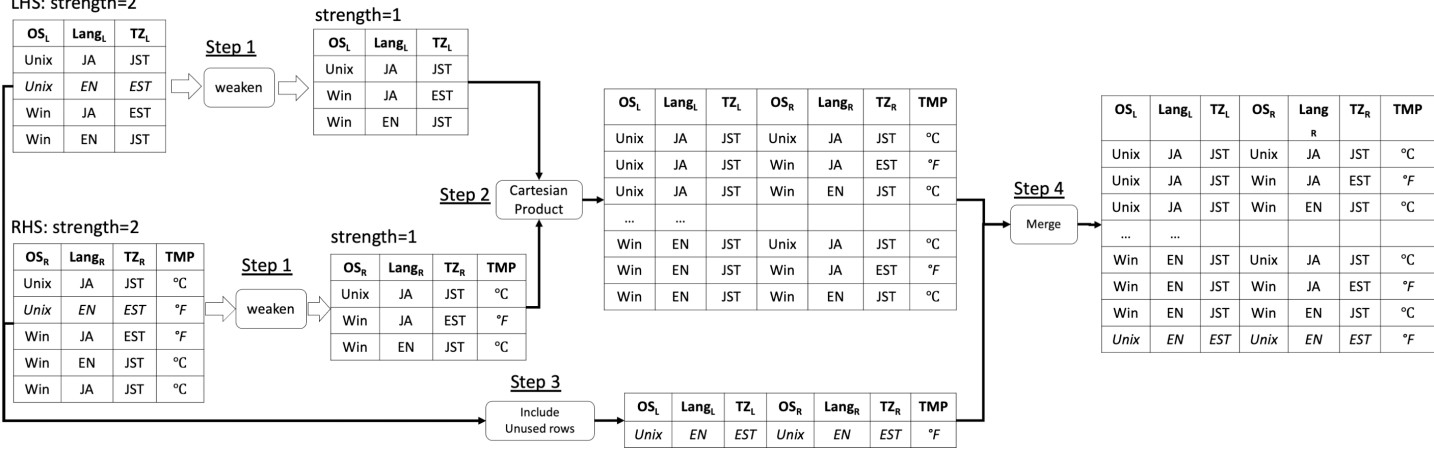

**Figure 1  Running example of weaken-product based combinatorial join.**

is an operation which can create a new covering array from already generated ones, we can utilize an optimal tool for each input.

We also expect it to accelerate the overall generation even with the overhead of combining smaller input covering arrays and enhance the applicability of CIT technique toward the larger and more complicated software products. In this section, we first illustrate the procedure of our proposed technique "weaken-product based combinatorial join" with a running example, which implements the "combinatorial join" operation. We next introduce some notations and a formal definition of this technique. After the formal definition of the technique, we define the operation "combinatorial join" in a more general way that allows other implementations of this operation, in addition to our "weaken-product based" method.

## A running example

We present a running example of our proposed algorithm weaken-product based combinatorial join with a concrete example (Fig. 1) where both the input arrays' and the output array's strength are $t = 2$. In this example, the original LHS is a covering array that contains three parameters (*i.e.*, $OS_L$, $Lang_L$, and $TZ_L$), each of which has two possible values Unix, Win, JA, EN, and EST, JST respectively. There is no constraint across LHS and RHS. Note that LHS and RHS can have different numbers of rows (*i.e.*, different sizes) and columns as shown in the diagram (Fig. 1). The original RHS is also a covering array that contains three parameters which are $OS_R$, $Lang_R$, and $TZ_R$ and they have the same possible values as the corresponding one in LHS. The goal of our algorithm (or method) is to combine them into one covering array that covers all the t-way combinations (in this example, $t = 2$) across the LHS and the RHS arrays without creating a new row neither in LHS nor RHS part.

First, the *weaken* operation, which shrinks the input covering array into another one with lower strength, is executed for both LHS and RHS (Step 1). The operation can have only one output. In general, the output arrays of this step in LHS will be covering arrays

with strength $t-1$, $t-2$, …, 1, while the corresponding arrays from RHS will be 1, 2, …, $t-1$. In this example, after this step, the output of LHS is only one covering array with strength 1 because the strength of the original LHS is $t=2$, and the output of RHS is also one covering array whose strength is 1. Next, for each pair of output arrays of Step 1, a *Cartesian Product* is performed and the results are merged into one (Step 2). As it is seen in the figure, for each row in the output of Step 1 from LHS, every row in the output of Step 1 from RHS is connected. For instance, for a row $(Unix, JA, JST)$ in LHS, every row in the output of the *weaken* operation for RHS $(Unix, JA, JST), (Win, JA, EST), (Win, EN, JST)$ is associated.

In this step, rows in the output with exactly the same values for all parameters are removed. This removal is necessary when the *weaken − product* is performed for the strength higher than 2 because the Step 1 is repeated multiple times and it may generate duplicated rows in the output.

Then, the remaining rows in LHS and RHS that do not appear in the output of Step 2 are connected and included in the final output in (Step 3). For example, the row $(Unix, EN, EST)$ in LHS and RHS is not found in the output of Step 2 and unless Step 3 is done to make up the missing tuples, not all the t-way combinations inside the LHS and RHS are ensured to be covered. Step 2 guarantees that $t$-way combinations of parameter values across LHS and RHS are covered. Step 3 guarantees $t$-way combinations of parameters inside LHS and RHS are covered. Therefore, the entire output becomes a covering array of strength $t$. Finally, the rows generated in Step 2 and Step 3 are merged into one array (Step 4).

## Notation

Now we define some notations in order to formalize our proposed method "weaken-product based combinatorial join" in Method of "weaken-product based combinatorial join". We first introduce a set of necessary functions before describing our proposed function, $weaken\_product(LHS, RHS, t)$ that builds a new covering array from two input arrays. The function takes three parameters, $LHS$, $RHS$, and $t$. The output of the function is an array containing all the factors held by the input arrays. $LHS$ and $RHS$ are arrays that do not have the same factors in common. In general, they are covering arrays of strength greater than $t$, although this condition is not mandatory. For simplicity, we assume that $LHS$ and $RHS$ do not have any constraints inside them. However, the proposed mechanism can handle those under constraints transparently. If the input has higher strength, it will be kept in the output, too, and if its rows do not violate given constraints, rows in output will also not violate the constraints. This is given as

$$weaken(A, i) = A_w \tag{2}$$

where *weaken* is a function that returns a new array from input $A$. The output has the following features:

- It has all the factors in $A$ and only those factors.
- It contains all the tuple of strength $i$ that appear in $A$.
- It contains rows that appear in $A$ and only those.

- Each row in the array is unique.

When output of the *weaken*$(A, i)$ is constructed, depending on the order of selecting rows from $A$, the size of the output can be different. Our implementation chooses to select a row that contains the most key-value pairs that are not covered in the output so far.

In the case the input $A$ is a covering array of strength $i$ or greater, *weaken*$(A, i)$ will be a covering array of strength $i$ and its size can be smaller than $A$. This is expressed as

$$|weaken(A, i)| \leq |A| \tag{3}$$

*factors* is a function that returns a set of factors on which a given array is constructed.

$$factors(A) = F \tag{4}$$

$F$ is a set of all the factors that appear in an array A

*project*$(A, f)$ is a function that returns an array created from an input array $A$ and a set of factors $f$.

$$project(A, f) = P \tag{5}$$

The returned array $P$ satisfies the following characteristics.

- It has all the factors given by $f$ only.
- For each row in $P$, a row in $A$, which contains the row, can be be found.

*connect* is a function that returns an array created from a couple of given arrays, $L$ and $R$.

$$connect(L, R) = C \tag{6}$$

The returned array satisfies following the characteristics.

- It has all the factors that appear in $L$ and $R$.
- *project*$(C, factors(L))$ contains all the rows found in $L$ and all the rows in it are contained by $L$.
- *project*$(C, factors(R))$ contains all the rows found in $R$ and all the rows in it are contained by $R$.
- Each row in $C$ has values for all the factors from $L$ and $R$.
- Each row in the array is unique.

Since there is not a requirement for combinations of rows from $A$ and $B$, $|C|$ can be as small as $\max(L, R)$.

$$set(A) = S \tag{7}$$

$S$ is a set that contains all the identical rows in an array $A$.

## Method of "weaken-product based combinatorial join"

Based on the formulae in 'Notation', the operation we propose *weaken_product* can be defined as follows.

$$
\begin{aligned}
WP &= weaken\_product(LHS, RHS, t) \\
&= [\bigcup_{i=1}^{t} weaken(LHS, i) \times weaken(RHS, t-i)] \cup connect(LHS_{unused}, RHS_{unused})
\end{aligned}
\tag{8}
$$

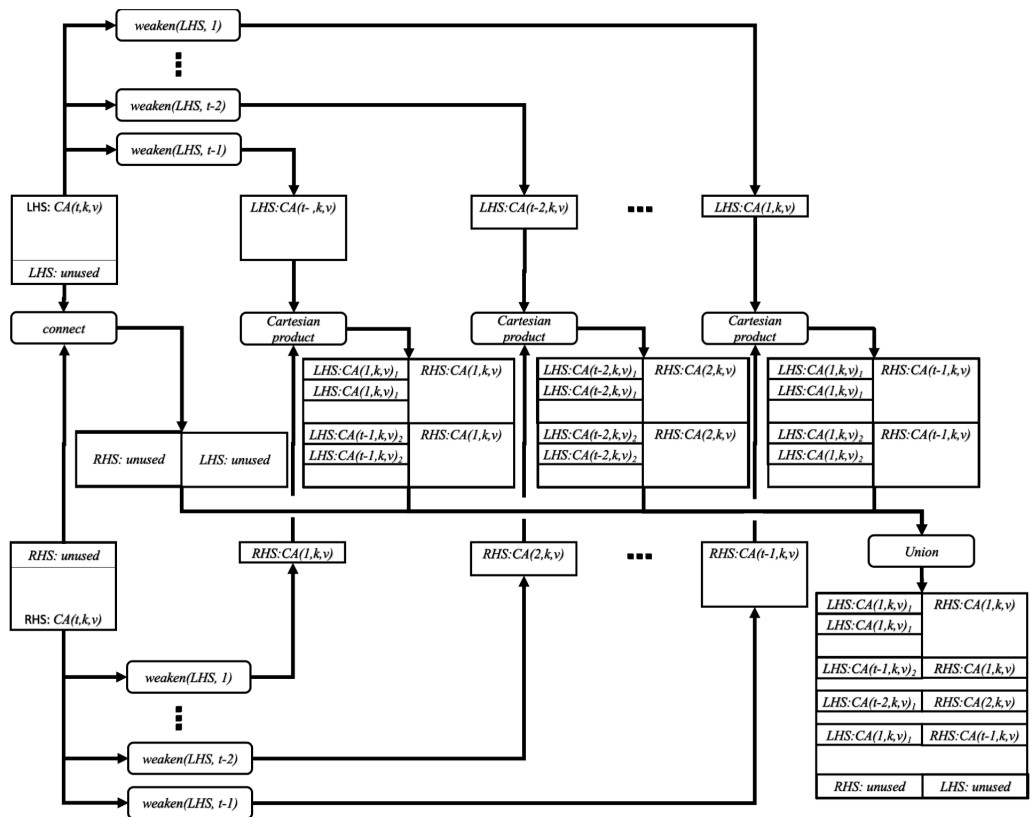

**Figure 2** Joining two covering arrays by weaken-product based combinatorial join.

where

$$LHS_{unused} = LHS \setminus project(W, factors(LHS))$$
$$RHS_{unused} = RHS \setminus project(W, factors(RHS)) \qquad (9)$$
$$W = weaken\_product(LHS, RHS, i)$$

Figure 2 illustrates the idea of the *weaken_product* function.

Next, we describe the characteristics of the output arrays generated by our proposed algorithm, in order to explain why we can use our algorithm to combine covering arrays generated under constraints. Given a set of parameters with their possible values, as well as a set of $t - way$ tuples that is called "Forbidden tuples", an array that covers all the possible $t$-way tuples but the forbidden ones is called a "constrained covering array" or CCA (*Cohen, Dwyer & Shi, 2008*). The set of forbidden tuples are determined by the constraints under which a covering array is generated for the system under test.

Suppose that *LHS* and *RHS* are constrained covering arrays generated under constraints with strength $t$. All rows in *LHS* are ensured to exist in *WP* and no new row is introduced according to Eqs. (2) and (8) ken,eq:weaken_product. This is also true for *RHS*. This leads to Theorem 1.

**Theorem 1**

$$set(project(WP, factors(LHS))) = set(LHS) \tag{10}$$

$$set(project(WP, factors(RHS))) = set(RHS) \tag{11}$$

We demonstrate that *WP* is a *CCA* generated under the constraints of *LHS* and *RHS*. From the precondition of the operation, there is no constraint across *LHS* and *RHS*. It is clear that there is no row that violates given constraints in *WP*. A tuple $T$ ($|T| = t$) that should be covered by *WP*, can be categorized into three.

- A tuple inside *LHS* (Eq. (10)).
- A tuple inside *RHS* (Eq. (11)).
- A tuple across *LHS* and *RHS*.

All the tuples that should be covered by *WP* inside *LHS* and *RHS* are found in the array (Theorem 1). In order to guarantee all the tuples across *LHS* and *RHS* are found in the *WP*, it is sufficient to include:

$$weaken(LHS, i) \times weaken(RHS, t - i) \tag{12}$$

where $0 < i < t$. Those are guaranteed to be in *WP* by the definition of the *weaken − product* operation defined as Eq. (8).

Thus, we can construct a new CCA from the existing CCA's without inspecting into neither the semantics of the constraints nor the forbidden tuples defined for the input arrays. This allows users to employ an approach, where different CIT tools to construct input covering arrays and then combine them into one, later.

The same discussion holds for constructing VSCA, when input arrays are the covering arrays of the higher strength than $t$.

## General definition of combinatorial join

We can generalize the operation we discussed in a way where our proposed method and *Ukai et al. (2019)* can be considered as implementations of one abstract operation based on the ideas introduced in 'Notation'. This improves the approach in our last work. The characteristics that are desired for the output of the operation can be described as follows.

$$set(project(combinatorial\_join(LHS, RHS, t), factors(LHS)) = set(LHS)) \tag{13}$$

$$set(project(combinatorial\_join(LHS, RHS, t), factors(RHS)) = set(RHS)) \tag{14}$$

$$\begin{aligned} tuples(project(combinatorial\_join(LHS, RHS, t), t)) \supset \\ tuples(LHS \times RHS, t) \setminus tuples(LHS, t) \cup tuples(RHS, t) \end{aligned} \tag{15}$$

where $tuples(A, t)$ is a function that returns a set of all the t-way tuples in an array $A$.

In this definition, note that any requirements are not placed on the input arrays. They do not need to be even any sort of covering arrays. These characteristics ensure that the operation does not introduce a new row that may violate constraints given to *LHS* or *RHS* and that it covers all the possible $t$-way tuples in and across *LHS* and *RHS*.

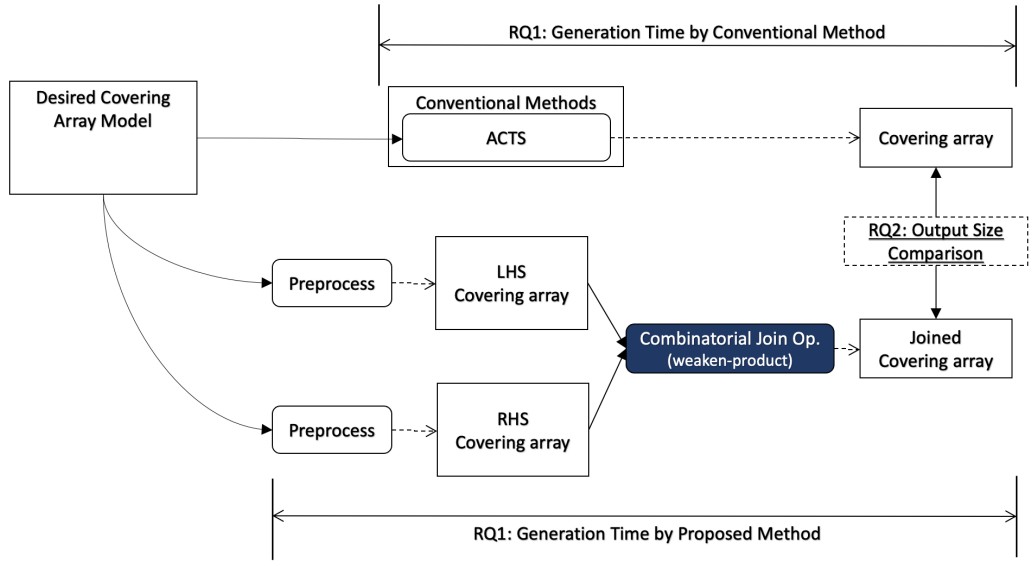

**Figure 3** Research questions (overview).

## EVALUATION

### Research questions

In order to evaluate our technique from the aforementioned perspectives, we are going to answer the following research questions:

- **RQ1: Can our weaken-product combinatorial join technique accelerate the existing CIT tools in covering array generation?**
- **RQ2: How are the sizes of covering arrays generated through our combinatorial join technique compared to the sizes of covering arrays generated without it?**
- **RQ3: Can our approach reuse test oracles?**
- **RQ4: How can our approach handle constraints with flexibility?**

There is another approach that constructs a new covering array from existing ones (*Zamansky et al., 2017*). However it relies on converting an input array into a factor by reckoning each row in it as a level of the factor. This approach is not practical unless the number of factors are small. Due to the scalability issue, it is inapplicable to the experiment subjects used in our study. Hence, we are not going to compare our approach's performance with their method but with that of ACTS.

### Evaluation methodology

In this section, we describe how we conduct evaluation to answer each research question, and we illustrate how each research question relates to the covering array generation process in Fig. 3.

In order to answer RQ1, we measure the execution time of our algorithm including necessary preprocesses for the input data for a desired input model. The preprocess may contain a covering array generation since our algorithm does not generate a covering array

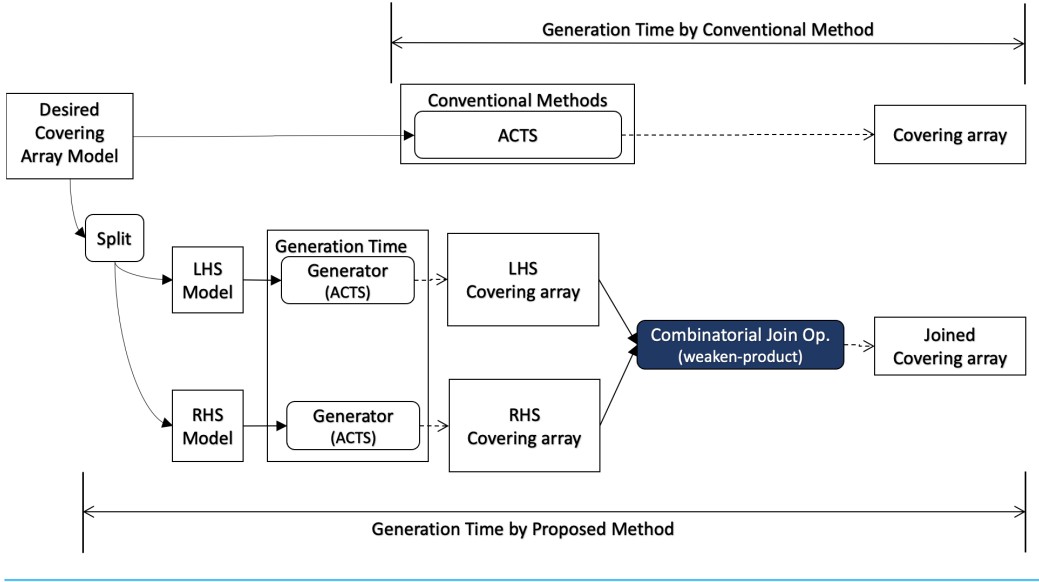

**Figure 4  Scratch generation.**

but it takes two covering arrays as input. It will be compared with the execution time to generate a covering array using a conventional method for the same desired output model.

In order to generate covering arrays in our experiments, we need an external tool that executes the process and we chose ACTS for it. The reason why we chose ACTS is because it is not only widely used but also the fastest one among the tools available for us. We considered PICT as another choice, however it turned out to be too slow for our experiments because of its specification, where its covering array construction with constraint handling requires exponential time along with the number of factors (*Czerwonka, 2016*).

Similarly, the sizes of the generated covering arrays by the proposed method and conventional method are compared (RQ2).

When covering array generation is executed from scratch, the preprocess for the desired covering array model consists of two parts as illustrated in Fig. 4. One is to split the mode into LHS and RHS and the other is to generate covering arrays for them respectively. For splitting the model, we can think of some strategies. One is to divide the input into two groups each of which has the same number of factors.

Moreover, well-known covering array generation tools support a feature called "seeds" or "incremental generation", where an existing covering array is given as input whose rows are ensured to appear in output. This feature enables users to reuse test cases, test results, test oracles, *etc.* along with the input covering array. In this scenario (Fig. 5), the requirements for the final output ("Desired Covering Array Model" in the diagram) and base covering array for the conventional method are given as input. On the other hand, for our method, the factors to be added to the seeds are given separately ("RHS Model" in the

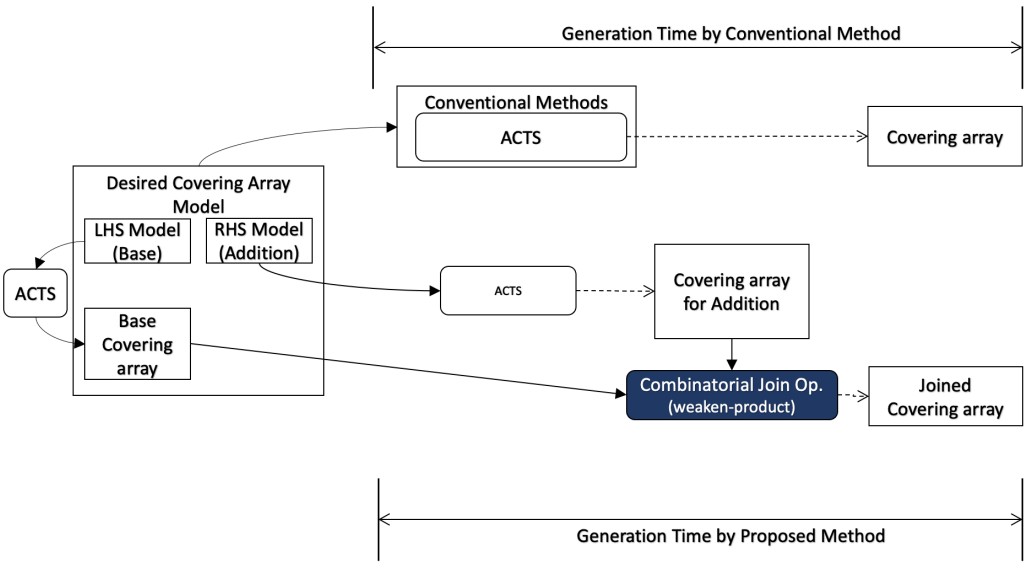

**Figure 5** **Incremental generation.**

diagram) and it is necessary to take into account the time to generate a covering array for it. However, the base covering array can be used as LHS without any preprocessing.

Our approach constructs a new row by selecting rows from input arrays instead of constructing it from scratch, so it has less options to optimize (minimize) the size of its output. As a result, our approach cannot generate a smaller output array than the conventional method (RQ2). In order to answer RQ2, we will compare the size of covering arrays generated by our method and the conventional method.

Those comparisons are conducted for artificial models designed based on our experience and well-known models distributed as Real-world benchmark (*Cai, 2020*). This benchmark contains six real-world instances. They are extracted from real test suites for Apache, Bugzilla, GCC, *etc.* Those models have from 20 up to 1,000 degrees with constraints.

Our approach allows us to reuse test cases defined as input covering arrays, but the reusability of test oracles along with the input covering arrays is an independent question. In order to answer RQ3, we will extend our previous work (*Ukai et al., 2019*) by examining various scenarios where test oracles may be reusable or not.

Since constraint handling in CIT is an area actively being studied, there are a number of techniques each of which has its own pros-and-cons in performance, flexibility, and other aspects. Hence, it is beneficial to apply "divide-and-conquer" approach to generation of a covering array so that we can utilize multiple covering array generators in combination. We will answer RQ4 by examining the detail of the procedure to employ the technique to implement the approach.

### Independent variables

As mentioned already, we measure the generation time and size of output covering arrays (the **dependent variables** of our evaluation), for various set of settings along with different

number of parameters. One suite of settings is characterized by *G* eneration Scenario and *D* esired Covering Array Model, which usually consists of *D* egree, *R* ank, *S* trength, and *C* onstraint Set. We describe each of there **independent variables** in our evaluation in the next sections.

### Generation scenario

We define a couple of scenarios to generate a covering array using our weaken-product based combinatorial join approach:

1. Generating a covering array from scratch;
2. Generating a covering array incrementally.

The first one refers to a scenario, in which a covering array is generated from a couple of given models from scratch. In this scenario, we expect that our approach can improve the overall generation time by executing a CIT tool concurrently and then combining the arrays generated in parallel. Especially, we expect our approach accelerates the generation of a covering array with a large number of parameters in higher strength or under complex constraints. Because in such situations, the generation time grows more rapid than linear and the approach makes it possible to apply "divide-and-conquer" to build the final output covering array. To maximize the improvement, we use the same model for generating both *LHS* (left hand side) and *RHS* (right hand side) covering arrays, because in this case the input arrays for the join operation are generated in the same amount of time.

In the second scenario, a new covering array with the specified degree and constraint set is generated from an existing covering array. Incremental generation is useful when, for instance, there is already a covering-array-based test suite for a certain component and a regression test is required for this component because new attributes are added to it. In this use case, there is already a test suite (a covering array) whose test oracles are defined. By employing incremental join, we do not need to define test oracles for a completely new covering array. In this scenario, we expect our approach to accelerate the generation time because our approach does not require to re-calculate tuples to be covered by the input arrays.

### Strength of the output covering array

*Strength* is the overall combinatorial coverage guaranteed in the output. In our experiments, we use 2 and 3 because higher strength covering array generation in this degree is not practical since both of ACTS and our *weaken−product* algorithm were too much time consuming.

We can also think of a covering array some of whose factors can be considered a higher strength covering array, which is called a variable strength covering array (VSCA). By employing weaken-product based combinatorial join, we can think of a method to construct a VSCA. That is, if we give a couple of covering arrays each of whose strength is 3 or higher and perform a combinatorial join operation with strength 2, the operation results in a new VSCA. For VSCAs, we only conduct the scratch generation experiments and the output covering array consists of two sub-covering arrays of a higher strength (3 or 4) and the same degree.

The second one, real-world benchmark models, we use the original factors and constraints as they are provided. The factors are split into two groups of factors, which are referenced by a constraint at least once and which are not referenced by any constraints.

### Input parameter models

We used two types of input parameter models to generate covering arrays in our evaluation. One is "synthetic" parameter models and the other is "real-world" benchmarks. For the synthetic models, the total number of parameters (factors), which is the degree of a model, ranges from 20 up to 980 in strength 2. In the strength 3, it will be moved from 20 up to 380. Each parameter has four possible values (the levels of each parameter). When the scratch generation scenario is performed, the LHS and the RHS are defined to have the same size. For example, if we are going to generate a covering array whose degree is 500, both LHS and RHS will be set with 250 parameters. This rule is also applied for the VSCA generation scenario. The number of parameters will be moved from 20 to 380 when a VSCA ($t = 2,3$) is generated, while it will be moved from 20 to 80 when a VSCA ($t = 2,4$) is generated.

For the incremental generation scenario the RHS is always set to 10 and the rest is assigned to the LHS. For example, when the total number of the parameters is 500, the LHS will have 490 parameters and the RHS will have 10 parameters. This design of model is based on the consideration that incremental generation is useful when you want to reuse test oracles defined for the initial covering arrays (*i.e.*, the LHS) and the benefit is more remarkable when the existing test suite (*i.e.*, the LHS) for a system under test is large, in which case the reusable objects (*i.e.*, test oracles) is plentiful, while the number of parameters added to the system are relatively smaller (*i.e.*, the RHS), which requires new creation of test oracles.

The other type of models is "real-world" parameter models. We used "CASA" benchmark models, which is widely referenced in CIT area, in our evaluation (*Cai, 2020*). It includes various sets of parameter models taken from real world projects and we selected the following data sets for our evaluation.

- APACHE (172 factors, 2–4 levels, 7 predicates)
- BUGZILLA (51 factors, 2 levels, 5 predicates)
- GCC (199 factors, 2 levels, 40 predicates)
- SPINS (18 factors, 2–4 levels, 13 predicates)
- SPINV (55 factors, 2–4 levels, 49 predicates)
- TCAS (12 factors, 2–10 levels, 3 predicates)

The largest one is GCC and it has 199 parameters while the smallest is TCAS and it has 12 parameters. Each data set has its own constraint set. The GCC model has a constraint set which consists of 40 predicates for instance. For the real-world parameter models, only the scratch generation scenario is performed. The parameters involved in any constraints are grouped into the LHS side and parameters not involved in any constraints are grouped into the RHS side.

[1] This constraint can be simplified by manual transformation. However ACTS does not perform such a transformation by itself.

### Constraint set

In our evaluation for the synthetic models, three *constraintsets* are defined and used, which are none, basic, and basic+.

There are real world practices that generate a combinatorial test suite from a high-level model such as a regular expression or a finite state machine (*Usaola et al., 2017*; *Bombarda & Gargantini, 2020*). Such high-level input models are turned into large parameter models with complex constraint sets and then they are processed by CIT tools, hence it's hard to find any good benchmark factor-constraint sets for such models. In order to simulate this situation, we expand and use a software model originally designed to evaluate ACTS (*Kuhn, Kacker & Lei, 2008*; *Yu et al., 2013*; *Computer Security Research Center, 2016*) by designing and generating various constraint sets for it.

The original model had only ten factors, we expand it by repeating the same factors and constraint set $n$ times.

In order to observe how dependent variable behave when a different set of constraints is given. The value "none" means no constraint was specified on a covering array generation. If the value "basic" is specified, a set of constraint defined by a following Eq. (16) is used.

$$p_{10i+1} > p_{10i+2} \vee p_{10i+3} > p_{10i+4} \vee p_{10i+5} > p_{10i+6} \vee p_{10i+7} > p_{10i+8} \vee p_{10i+9} > p_{10i+2}$$
$$(0 \leq i < n) \tag{16}$$

$n$ is a variable, which is used to control the number of degrees in an experiment. The other constraint set is defined as follows.

$$(p_{10i+1} > p_{10i+2} \vee p_{10i+3} > p_{10i+4} \vee p_{10i+5} > p_{10i+6} \vee p_{10i+7} > p_{10i+8} \vee p_{10i+9} > p_{10i+2})$$
$$\wedge p_{10i+10} > p_{10i+1}$$
$$\wedge p_{10i+9} > p_{10i+2}$$
$$\wedge p_{10i+8} > p_{10i+3} \qquad (17)$$
$$\wedge p_{10i+7} > p_{10i+4}$$
$$\wedge p_{10i+6} > p_{10i+5}$$
$$(0 \leq i < n)$$

This was designed by adding several conditions to the "basic" set and made more complex than it in order to understand how covering array generation is affected by complexity of given constraints[1].

## RESULTS

In this section, we present and discuss the results of our evaluation. All the experiments in this section are executed on the computer with Intel(R) Core(TM) i9 2.40 GHz (8 cores) CPU and 32GB memory working on macOS Catalina Version 10.15.7.

### Covering array generation time
#### Scratch Generation

Figures 6, 7 and 8 show the results of comparing the generation time between the covering arrays generated by our method and ACTS, given the strength set to 2 and the degree set

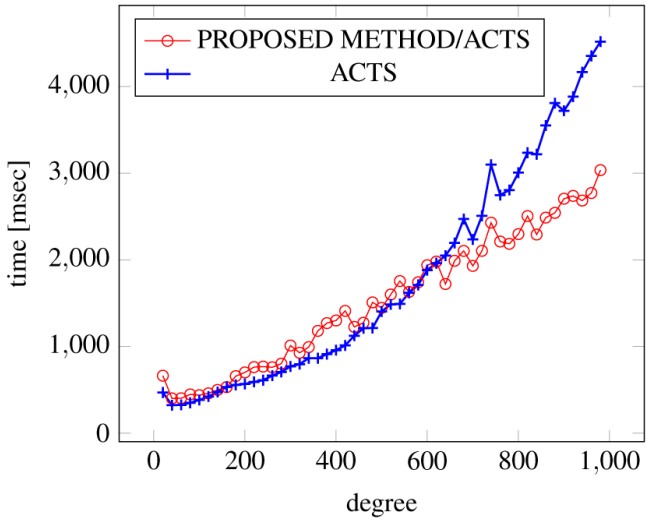

**Figure 6** Scratch generation; $t = 2$; constraint = none.

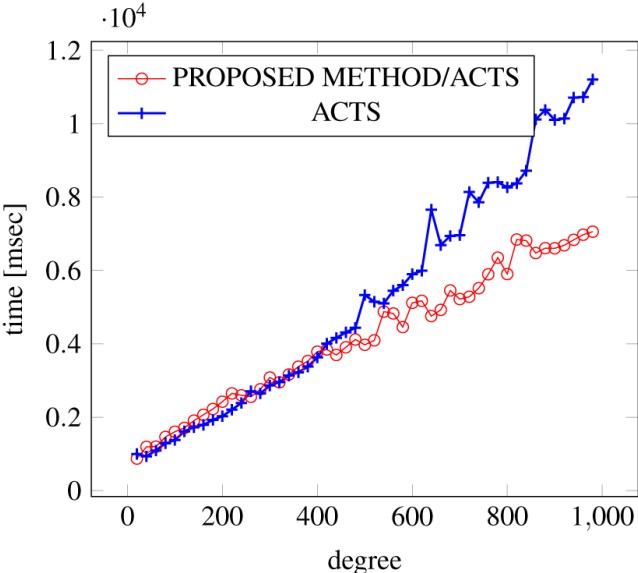

**Figure 7** Scratch generation; $t = 2$; constraint = basic.

up to 1,000, as it represents a large scale industrial system specification (Strength of the output covering array).

As shown in the figures, as the degree increases, our approach reduces the generation more remarkably. When the strength is 2 and degree is 980, the time is reduced by 21% to 25% or more, with or without constraint sets.

Figures 9, 10 and 11 show the results of comparing the generation time between the covering arrays generated by our method and ACTS, given the strength set to 3 and the

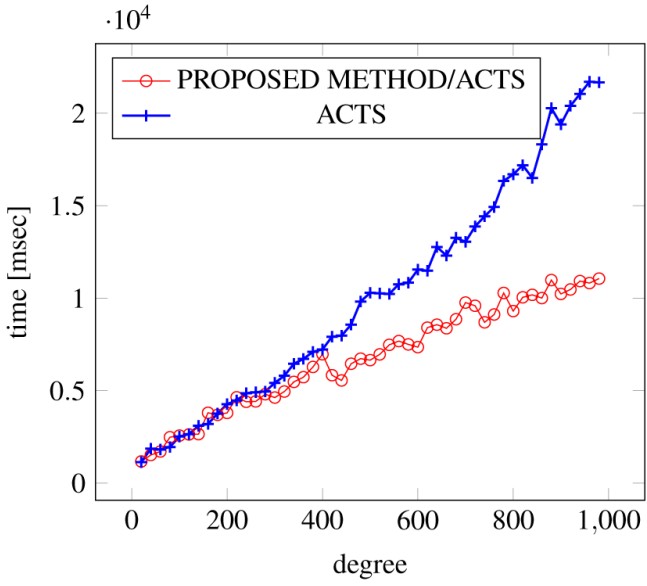

**Figure 8** Scratch Generation; $t = 2$; constraint = basic+).

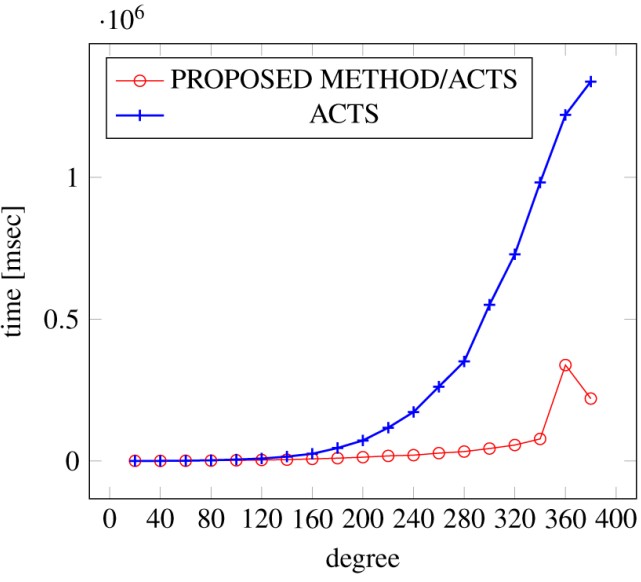

**Figure 9** Scratch generation; $t = 3$; constraint = none.

degree set up to 380. With strength 3, we see greater time efficiency improvements than with strength 2, because the covering array generation time grows more rapidly along with the increase of degree, when a higher strength is specified. Specifically, our approach reduces the generation time by 89% to 91% compared to ACTS, when the strength is 3 and degree is 380. Also, the generation time grows more rapidly when a more complex constraint set is specified. Thus, in the scratch generation scenario, we observe the greatest

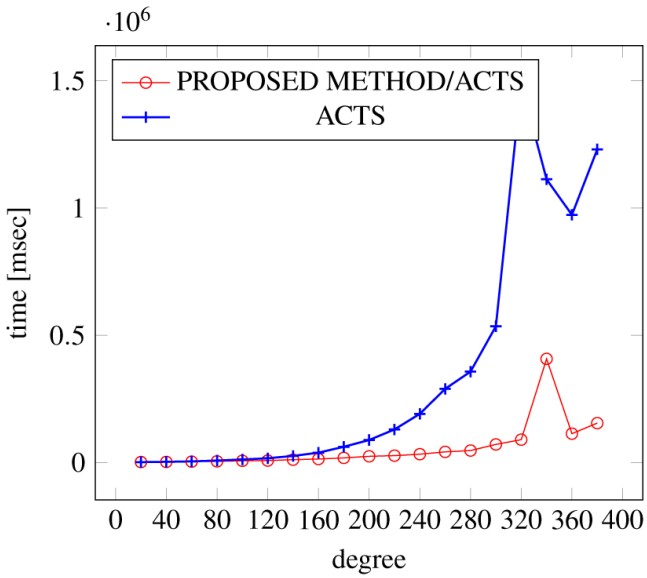

**Figure 10** Scratch generation; $t = 3$; constraint = basic.

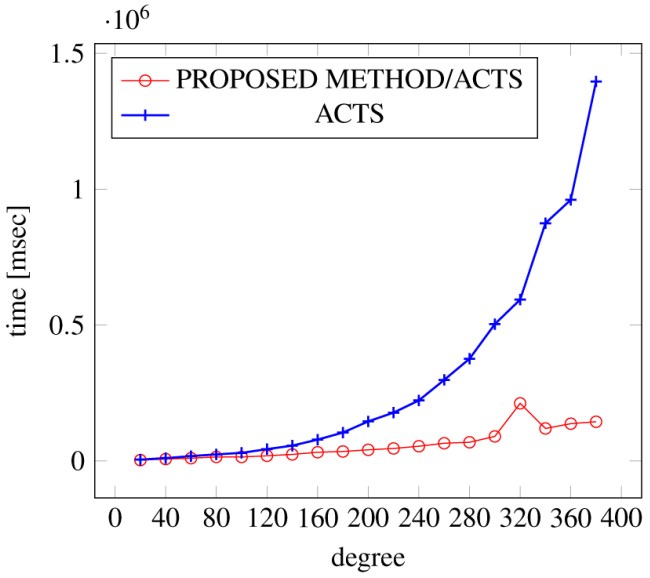

**Figure 11** Scratch generation; $t = 3$; constraint = basic+).

improvement when a strength is set to 3 ($t = 3$) and the basic+ constraint set is present among the settings.

### Variable strength covering array generation scenario

Figures 12, 13 and 14 show the results of comparing the VSCA ($t = 2, 3$) generation time between our method and ACTS, given a degree ranging from 20 to 380. Our approach

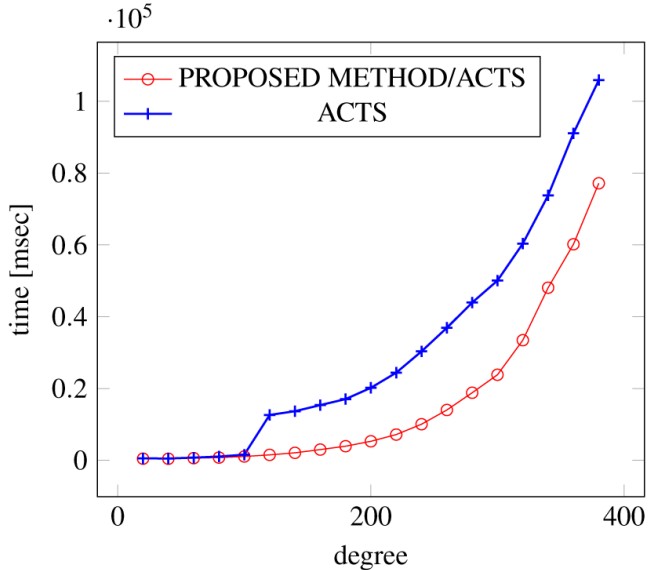

**Figure 12** VSCA generation; $t = 2$ and $t = 3$; constraint = none.

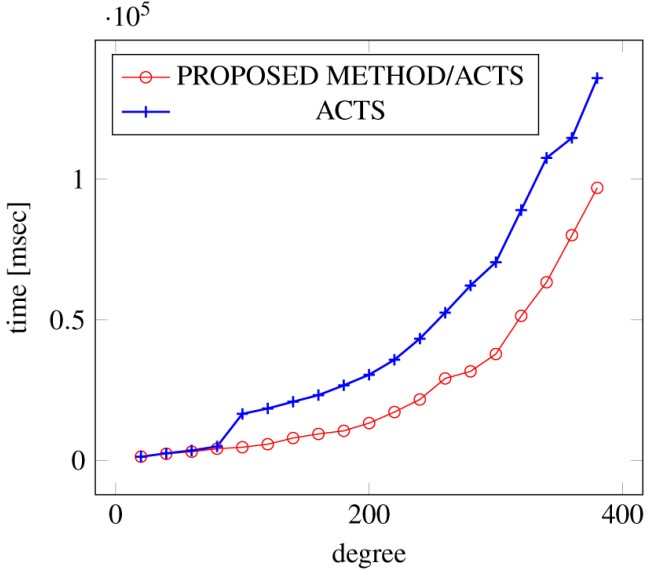

**Figure 13** VSCA generation; $t = 2$ and $t = 3$; constraint = basic.

reduces the generation time by 28%–30% compared to ACTS, when the mixed strengths are 2 and 3 and the degree is 380.

Figures 15, 16 and 17 show the results of comparing the VSCA ($t = 2, 4$) generation time between our method and ACTS, given a degree ranging from 20 to 160. Our approach

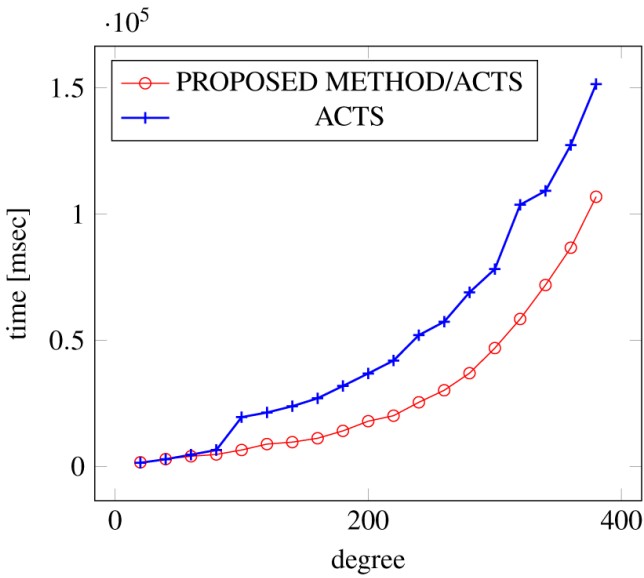

**Figure 14** VSCA generation; $t = 2$ and $t = 3$ (constraint = basic+).

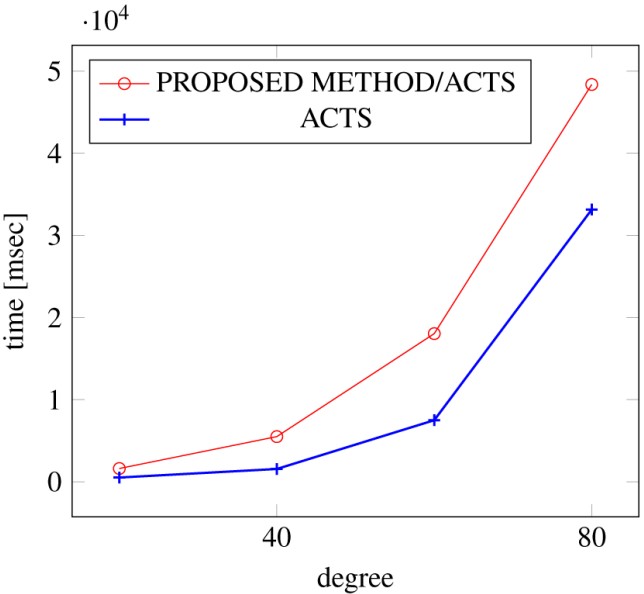

**Figure 15** VSCA generation; $t = 2$ and $t = 4$; constraint = none.

reduces the generation time by up to 34% compared to ACTS, when the mixed strengths are 2 and 4 and the degree is 80.

Similar to the scratch generation scenario given a single strength, the generation time improvements are more remarkable when a higher strength is specified and a more

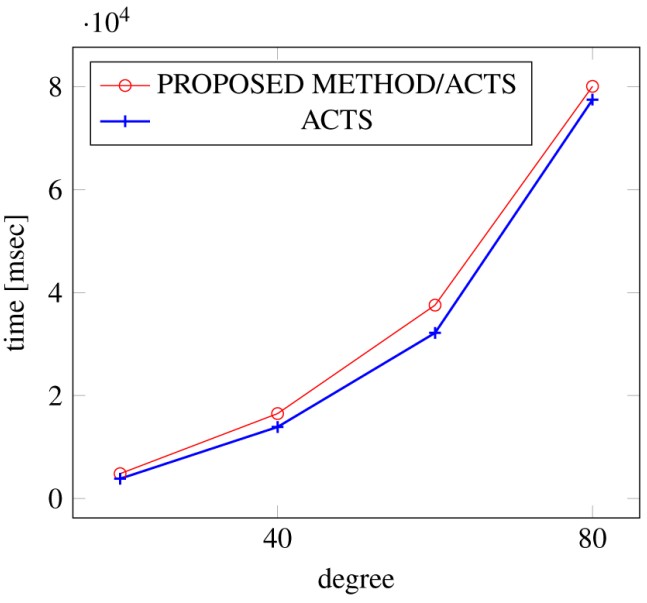

**Figure 16** VSCA generation; $t = 2$ and $t = 4$; constraint = basic.

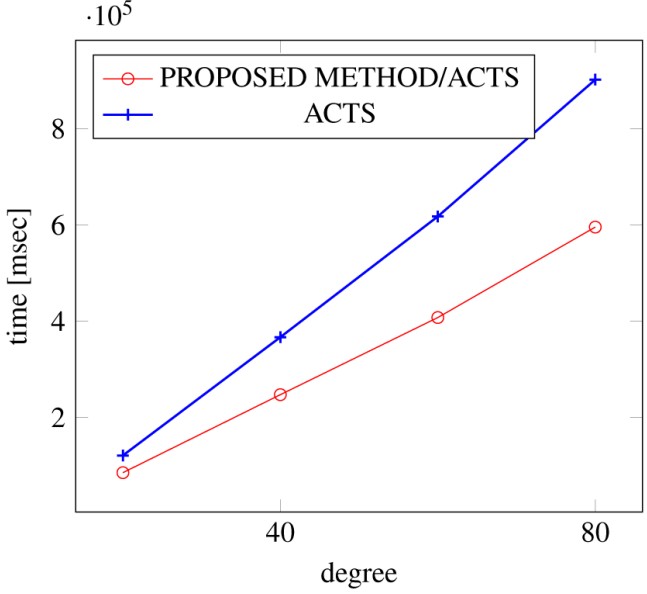

**Figure 17** VSCA generation; $t = 2$ and $t = 4$; constraint = basic+.

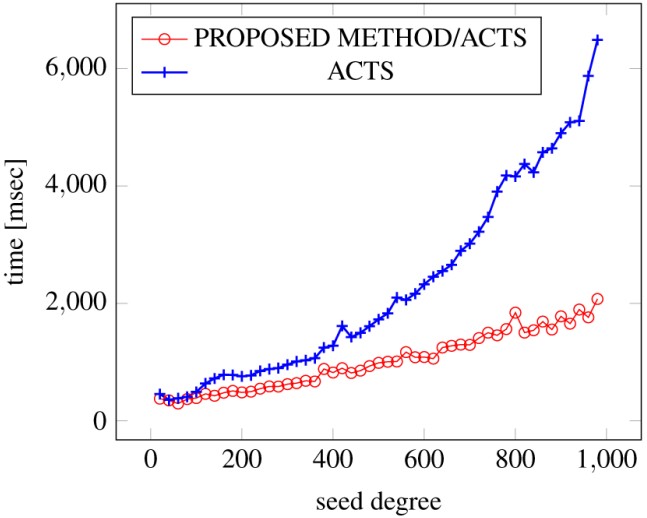

**Figure 18** Incremental generation; $t = 2$; constraint = none).

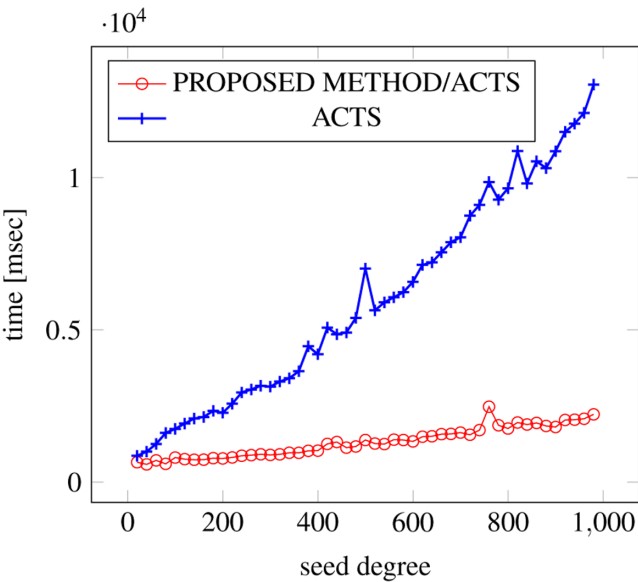

**Figure 19** Incremental generation; $t = 2$ (constraint = basic).

complex constraint set is given. However, the benefit is less significant for variable strength generation unless a complex constraint (basic+) set is given.

### Incremental generation scenario

Figures 18, 19 and 20 show the results of comparing the generation time between the covering arrays generated by our method and ACTS, given a degree set to 380 and the

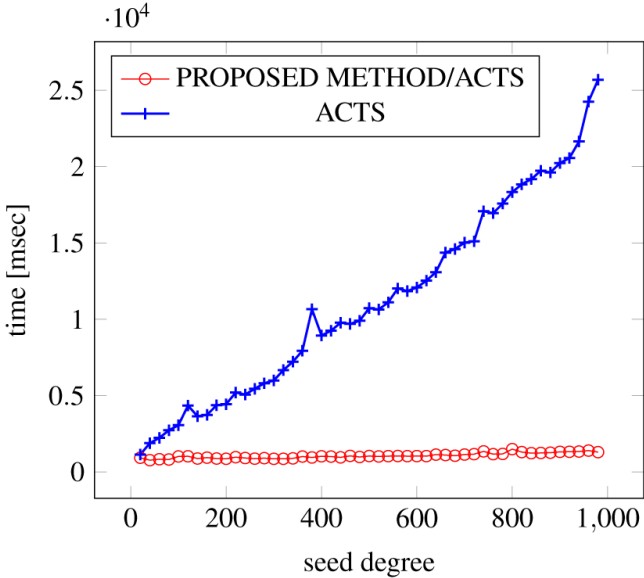

**Figure 20  Incremental generation; $t = 2$ (constraint = basic+).**

strength is 2. Our approach reduces the generation time by 84% to 98% compared to ACTS.

Similar to the scratch generation scenario, the greater generation time improvements are observed in the higher strength and with the more complex constraint sets. The improvements are more drastic than in the scratch generation scenario. This is because the conventional approach does not utilize the knowledge about the seed array which is already a covering array.

Figures 21, 22 and 23 show the results of comparing the generation time between the covering arrays generated by our method and ACTS, given a degree set to 380 and the strength is 3. In this case, our approach reduces the generation time by 99% compared to ACTS.

### Summary

RQ1: Can our weaken-product combinatorial join technique accelerate the existing CIT tools?

**Yes. When the degree is high (380–980), the acceleration (*i.e.*, the reduction of generation time) is significant. Specifically, in strength 2, our approach reduces the test suite generation time of synthetic systems by 33%–95%, while it reduces the generation time by 84%–99% in strength 3. For the VSCA generations, we observe 28%–34% generation time reduction.**

## Generated covering array size
### Scratch generation

Tables 1 and 2 show the sizes of generated covering arrays in *strength* 2 and 3 respectively. In strength 2, the degree of output covering array ranges from 20 to 980, and in strength 3,

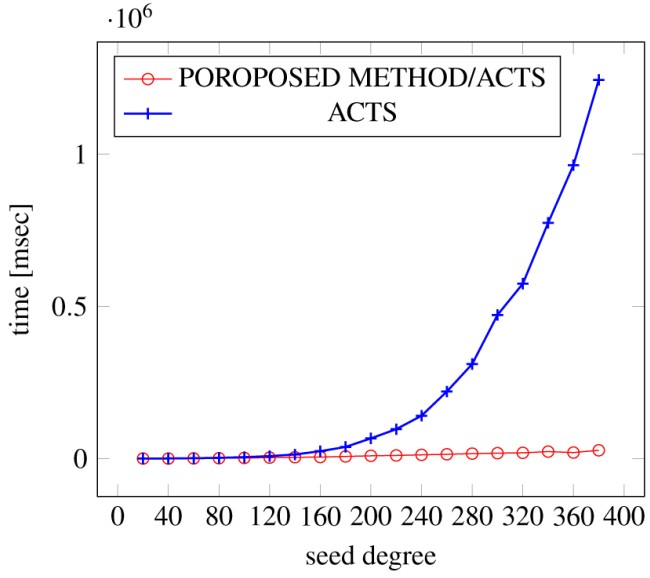

**Figure 21** Generation time; $t = 3$ (constraint = none).

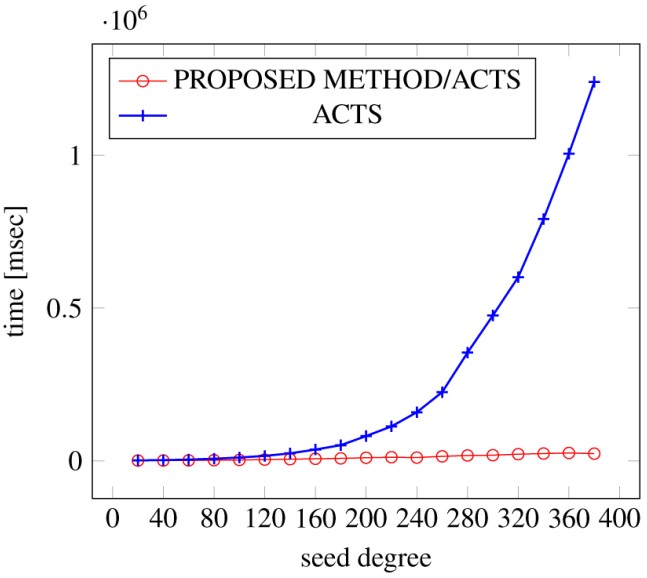

**Figure 22** Generation time; $t = 3$ (constraint = basic).

the degree ranges from 20 to 380. The "size penalty" represents the percentage the size is increased by our proposed method comparing to the conventional approach (ACTS). The size increase is named as a "penalty" for gaining a faster generation time.

As shown in Table 1, when the strength is set to 2, the size penalty varies from 35% to 90% depending on the constraint set at degree = 20, and it decreases to 30–43% when the degree is increased to 980. As shown in Table 2, in strength 3, the size penalty varies from

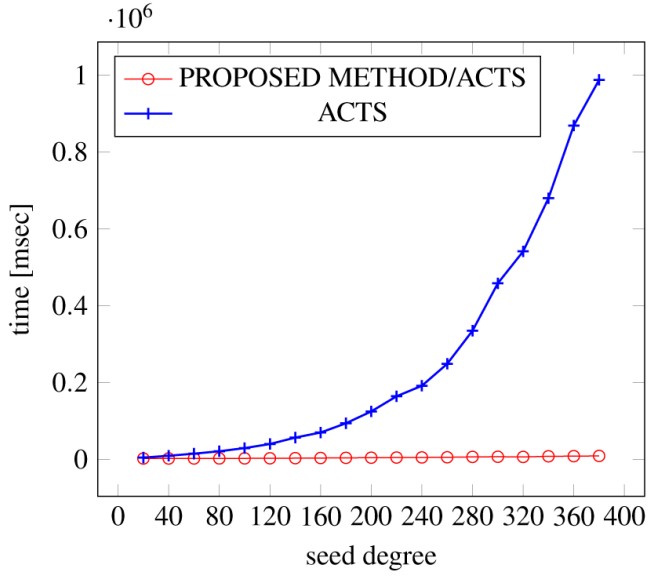

**Figure 23  Generation time; $t = 3$ (constraint = basic+).**

**Table 1  Size of covering arrays; scratch; $t = 2$; $d = [20, 980]$.**

| Constraint set | None | | Basic | | Basic+ | |
|---|---|---|---|---|---|---|
| | min | max | min | max | min | max |
| PROPOSED METHOD based on ACTS | 75 | 117 | 74 | 116 | 31 | 65 |
| ACTS | 41 | 82 | 39 | 80 | 23 | 50 |
| Size penalty with ACTS | 83% | 43% | 90% | 45% | 35% | 30% |

**Table 2  Size of covering arrays; scratch; $t = 3$; $d = [20, 380]$.**

| Constraint set | None | | Basic | | Basic+ | |
|---|---|---|---|---|---|---|
| | min | max | min | max | min | max |
| PROPOSED METHOD based on ACTS | 295 | 1356 | 455 | 1214 | 176 | 724 |
| ACTS | 208 | 562 | 228 | 567 | 118 | 301 |
| Size penalty with ACTS | 42% | 141% | 100% | 114% | 49% | 141% |

about 42% to 100% depending on the constraint set at degree = 20, and it increases to 114%–141% when the degree is increased to 980.

Greater size penalties were observed in $t = 3$ than in $t = 2$, but no clear correlation with the complexity of the constraint sets was seen. Our proposed method does not handle constraints by itself but let the underlying covering array generation tool (*i.e.*, ACTS) handle them, which our method is compared to. In other words, our method and ACTS handles constraints in the same way, so unless the size of the output from the underlying tool is impacted by the complexity of the constraint sets, we will not see the influence of the complexity of the constraint sets in the output sizes.

**Table 3** Size of covering arrays; VSCA; $t = 2, 3$; $d = [20, 380]$.

| Constraint set | None | | Basic | | Basic+ | |
|---|---|---|---|---|---|---|
| | min | max | min | max | min | max |
| PROPOSED METHOD based on ACTS | 163 | 330 | 191 | 339 | 88 | 176 |
| ACTS | 162 | 295 | 166 | 296 | 163 | 298 |
| Size penalty with ACTS | 0% | 8% | 15% | 8.8% | -46% | -43% |

**Table 4** Size of covering arrays; VSCA; $t = 2, 4$; $d = [20, 80]$.

| Constraint set | None | | Basic | | Basic+ | |
|---|---|---|---|---|---|---|
| | min | max | min | max | min | max |
| PROPOSED METHOD based on ACTS | 721 | 1763 | 773 | 1786 | 297 | 806 |
| ACTS | 760 | 1735 | 774 | 1742 | 290 | 785 |
| Size penalty with ACTS | −5% | 2% | 0% | 2.5% | 2.4% | 2.7% |

*Variable strength covering array generation scenario*

Tables 3 and 4 show the sizes of generated covering arrays in variable *strength* (2, 3) and (2, 4) respectively.

We constructed a VSCA by splitting all the factors into two groups with the same size, both of the groups have higher strength than 2 (*i.e.*, $t=3$ or 4) inside while the strength across the groups is 2.

When the VSCA's strengths are 2 (across a group) and 3 (inside a group), at the degree 20, the size penalty is −46–15% and it becomes −43–8.8% when the degree increases up to 380 (Table 3). When the VSCA's strengths are 2 and 4, we set an upper bound to the degree to 80 due to a long execution time (over 5 min) to construct such covering arrays by ACTS. It is not cost effective in constructing such experiments. At the degree 20, the size penalty is −5–2.5% and it increases up to 2–2.7% when the degree grows up to 80 (Table 4).

As seen in the tables, for the scratch generation, our method shows small size penalties or sometimes it even reduces the final output size. Such reductions in size are observed when a complicated constraint set (basic+) is given or a high strength (4) is given. It might suggest that ACTS is not optimized to generate VSCAs for such situations, but we were unable to identify the root cause. Unlike the conventional method which generates the entire VSCA all at once, our approach generates simple covering arrays with higher strength first and then connects them by our novel algorithm. This mechanism employed in our approach lets the covering array generation tool (*i.e.*, ACTS) leave out the consideration of covering tuples outside the original arrays, which may potentially increase both the output size and the generation time.

*Incremental generation scenario*

Tables 5 and 6 show the sizes of covering arrays generated by the incremental approach in *strength* 2 and 3 respectively. We ran experiments adjusting *LHS* (seeds) degree from 10 to 370 while the *RHS* degree is fixed to 10 for each setting.

**Table 5  Size of covering arrays; incremental; $t = 2$; $d = [20, 80]$.**

| Constraint set | None | | Basic | | Basic+ | |
|---|---|---|---|---|---|---|
| | min | max | min | max | min | max |
| PROPOSED METHOD based on ACTS | 75 | 124 | 74 | 122 | 31 | 65 |
| ACTS | 41 | 82 | 39 | 81 | 23 | 50 |
| Size penalty | 83% | 51% | 90% | 51% | 35% | 30% |

**Table 6  Size of covering arrays; incremental; $t = 3$; $d = [20, 380]$.**

| Constraint set | None | | Basic | | Basic+ | |
|---|---|---|---|---|---|---|
| | min | max | min | max | min | max |
| PROPOSED METHOD based on ACTS | 295 | 810 | 455 | 900 | 176 | 424 |
| ACTS | 208 | 562 | 228 | 567 | 118 | 301 |
| Size penalty | 41% | 44% | 100% | 60% | 49% | 41% |

In strength 2, the "size penalty" ranges from 35% to 90% at the *degree* $= 20$ while it decreases to 30%–51% with the degree increases (Table 5). In strength 3, the "size penalty" ranges from 41% to 100% at the *degree* $= 20$ while it decreases to 41%–49% when the degree increases to 380 (Table 6).

Similar to the scratch generation scenario, the larger size penalty is observed in the higher strength. But there is no clear relationship between the complexity of the constraint sets and the size penalty. The size penalty is less significant than in scratch generation scenario. This is because most of the output covering array is built by the underlying tool (ACTS) before our algorithm is performed and less additional rows are necessary to be added in our algorithm.

*Summary*

RQ2: How are the sizes of covering arrays generated by our combinatorial join technique compared to the sizes of covering arrays generated by the existing tools?

(1) In strength 2, our approach increases the size of output covering array (size penalty) by 35%–90%, and the size penalty becomes 41%–141% in strength 3, to generate a covering array from scratch or incrementally.(2) For VSCA generation, with strengths 2 and 3, the size penalty is −46%–15%. When the variable strengths are 2 and 4, it becomes −5%–2.7%.(3) The size penalty becomes smaller when more factors (or degrees) and more complex constraints are given.

## Reusability of test oracles by our method

Our previous work (*Ukai et al., 2019*) discussed how combinatorial join technique is employed to reuse test oracles over multiple software testing phases, in order to reduce total testing cost. The approach reuses the test oracles that are manually designed in the component level testing phase in later testing phases such as integration test, *etc.*, by applying combinatorial join. The results of the previous work show that combinatorial join can reduce overall testing cost by more than 55%, and the level of reduction depends on the complexity of the software under test (SUT) and the ratio of oracle designing cost

to test execution cost. However, there are several implicit assumptions behind that work, which we intend to study and discuss more in this paper, as follows:

1. Test oracles designed for one testing phase can be reused in the next testing phase.
   (a) Under what conditions the SUT should display the same behavior for the reused test input?
   (b) Under what conditions and what sort of bugs can be detected by reusing the test oracles in the SUT?
2. In a testing phase where these oracles are reused, none or a small amount of additional test oracles are required to be added.

We first examine these assumptions and further clarify the conditions where combinatorial join can reduce overall testing cost. For simplicity, in this discussion, we model the testing effort into two phases, "component level testing" and "system level testing". We then evaluate our method proposed in this paper based on those conditions.

The assumption is based on a couple of other underlying assumptions: first, with the same test oracles, new bugs can be detected in later phases when a component is integrated into the original system; second, the component for which the test oracles are designed should behave in the same way as it behaves before the integration.

In general, each component is designed as much independent of each other as possible (*i.e.*, low coupling), that is, with the minimal or none interaction between components, the integration of several component will not change the behavior of each single component. Therefore, as long as factors included in a test suite created for a certain component cover all the inputs that may affect the behaviour of that component, the oracles defined at the component level are also valid in the system level testing (1a). If a bug is detected in system level testing but not component level testing given the same test case (the same input values and the same test oracle), it means that some value combinations across multiple components are exercised in the system level testing, which is impossible to be detected inside a single component. In order to address the 1b, we can think of a few bug classes that would be detected by this approach in the system level testing, such as "resource conflict", "incorrect abstraction", and "unintended dependency". Next we explain each class with examples.

Specifically, "resource conflict" refers to a type of bug that is triggered by conflicting usage of resources shared among multiple components. A list of typical bug examples of this class is shown as follows.

- **Data Corruption:** A component modifies shared data (such as system configuration) in a way that other components do not expect, or a component removes a directory in which other components expect their data files to be placed, *etc.*
- **Out Of Resources:** A component consumes or occupies resources (*e.g.*, memory, disk space, network band width) more than it is allowed.
- **Dead Lock:** A component locks a resource (database table, file, shared memory, *etc.*), which other components try to access, but do not unlock it.

Oracles to detect the "Out Of Resources" and "Dead Lock" bugs are defined in a way agnostic to their input parameters. That is, for instance, an oracle for "Out Of

Resource'' may be described as ''An out of memory error should not be thrown during a test execution'', which does not require any re-design for new input parameters and will not introduce any additional cost anyway. Therefore, among the aforementioned examples of the ''resource conflict'' bug class, ''Data Corruption'' is the only type of bugs, which can be detected by reusing test oracles through our combinatorial join approach and in which case, our approach leads to a testing cost reduction, compared to the conventional method. A bug reported by *Yoonsik Park (2018)* is one instance in this class, where a bug that survived all unit tests for the Linux Kernel eventually caused data corruption in the QEMU virtual machine on the kernel.

The second class of bugs under consideration is ''Incorrect Abstraction''. A system sometimes has a component responsible for ''abstracting'' a lower level of components. For instance, a graphic card driver is such an abstracting component and a graphics card is an example of a lower level component. When another component is accessing system's graphical capability, it expects that the capability works transparently regardless of the type of graphics card and the settings of its performance parameters. However, when an application component that utilizes the graphics capability assumes a specific behavior for the abstracting component (graphics driver), but this specific behavior is only satisfied by specific implementations (a graphics card), this class of bugs may be observed. If the specification of the abstraction component is not sufficiently defined or the testing coverage over the abstraction component is insufficient, these bugs remain undetected until the system level testing. A real-world bug found for Ubuntu (Linux) and Nvidia graphics card combination that produced unintended noise belongs to this class (*Nvidia Corporation Corporation, 2019*) and it could be avoided if they had an appropriate test oracle for the input.

The third class of bugs were referred as ''unintended dependency''. Sometimes a component may unintentionally depend on an assumption that is sometimes broken when it is used as a part of the entire system. For instance, if a developer misses a system requirement that the product needs to be run not only on Linux but also on Windows and a file separator can only be ''/'', the product will break at the system level testing, if the ''OS'' component is integrated in the testing phase (because path names used in the product cannot be resolved correctly). Some real-word bugs are introduced by lack of such dependency considerations (*Netty Project Community, 2016*; *Kawaguchi, 2020*). These bugs could be detected if there were test oracles for normal functionality of the SUT (*i.e.*, checking if the Netty or Jenkins starts up and it responds to basic requests) and the test cases with these oracles were executed with a properly set-up configuration (*i.e.*, *installation= upgrade*, *OS= MicrosoftWindows*, *dotNetVersion*=4.0However, the parameters came from different components ( *installation* mode is a parameter of Jenkins and the *OS* and *dotNetVersion* are platform parameters) and only a specific combination can trigger the bug. This means just reusing oracles is not sufficient to detect them, but guaranteeing to cover combinations between parameters is also necessary. Our method enables both without resorting to Cartesian product between two covering arrays.

The assumption 2 is satisfied if there exists a component which faces a consumer of the entire system among the components under test, and a test suite for verifying the

**Table 7  Data types, constraint handlings, and covering array generation performance for $2^{100}$ by various CIT tools.**

|  | Types | Available operators | | | | Performance | |
|---|---|---|---|---|---|---|---|
|  |  | Comparison | Mathematical | Logical | Conditional | Size | Time |
| ACTS | bool, number, enum, range | <, <=, = | +, -, *, / | &&, \|\|, ! | Not Supported | 14 | <1.0 s |
| PICT | string, numeric | >, >=, <, <=, <>, = | Not supported | AND, OR, NOT | IF/THEN/ELSE | 15 | <1.0 s |
| JCUnit | All Java types | All Java operators | All Java operators | All Java operators | All Java operators | 18 | 6.5 s |

component can also be used as a test suite for verifying the entire system. This assumption holds when the system level testing only focuses on functionality, but this is not true in general. Instead, in practice, aspects that are not examined in earlier phases, need to be more focused in later or the last testing phase (*i.e.*, system testing), such as performance, availability, scalability, *etc.* Nevertheless, when a consumer facing component is present, our approach will at least reduce the cost of system level testing for the functionality aspects of the system.

In summary, reusing test oracles by our combinatorial join approach makes it possible to detect some classes of bugs in system level testing, which were not found in component level testing, without re-defining test oracles. These classes are "Data Corruption caused by Resource conflict", "Incorrect abstraction", and "Unintended dependencies between components". At the same time, by reusing test oracles, functionality testing cost can be reduced in system level testing.

RQ3: What benefits does reusing test oracles across testing phases by weaken-product based combinatorial join deliver and in what conditions?

**Reusing test oracles by combinatorial join can detect new bugs in system-level testing that are not found in earlier testing phases without extra manual effort. At the same time, by reusing test oracles, functionality testing cost can be reduced in system level testing.**

## Flexibility of weaken product combinatorial join

The combinatorial join operation produces a new covering array from two existing covering arrays, it does not create a new combination of values or handle constraints by itself. In other words, it does not matter how the existing arrays are generated. In our experiments so far, we only chose ACTS for generating the input arrays due to its popularity and high performance, but in actual use cases, any combinations of CIT tools can be utilized for the generation, depending on the actual requirements, characteristics and availability of tools, among other factors. Known CIT tools have different characteristics in performance (*i.e.*, generation time), size efficiencies and capabilities, as described in Table 7. As we can see from the table, each tool has its own strengths and weaknesses. We summarize them as follows.

- ACTS has the best efficiency in time and size.
- PICT provides more readable notation for defining data and constraints than ACTS.

- JCUnit has the highest capability in handling various data types and constraints and its notations are the most readable among the three tools. Some of its capabilities (*e.g.*, defining a constraint using a regular expression) cannot be replaced with neither ACTS nor PICT.

Given these characteristics, an optimal approach to build a covering array $C$ from a complex test model (or specification) is proposed as follows, by generating sub-covering arrays first:

1. Generate a covering array $A$ using ACTS for factors in the specification that are with constraints and can be implemented easily and directly in ACTS, or for factors that are without any constraint.
2. Generate a covering array $B$ using JCUnit for factors in the specification that are with complex constraints that cannot be implemented directly in ACTS.
3. Combine covering arrays $A$ and $B$ using the combinatorial join operation.

This approach enhances the applicability of CIT in use cases where any single tool cannot generate an appropriate test suite easily, efficiently, or even possibly. For instance, if an SUT has specification that involves too complex constraints for ACTS and/or too many factors for JCUnit to generate a single covering array (test suite) all at once, this proposed approach makes it possible to use the CIT methodology for testing such SUT.

In summary, the combinatorial join operation is agnostic to how input arrays are generated and therefore it makes possible to combine multiple methods/tools to build one covering array. As shown in this discussion, there are various tools each of which has its distinct pros and cons and it is beneficial to employ the combinatorial join technique to combine covering arrays built by different tools.

RQ4: How can our approach handle constraints with flexibility?

**The combinatorial join approach enables to build a covering array for a test model with numerous parameters and complex constraints using multiple CIT tools in combination, by taking full advantage of the strength of each tool, such as the high performance of ACTS and the rich constraint handling support of JCUnit.**

## Performance in various scenarios

In addition to the study with common settings, in this section, we study the proposed method's performance in time and size with different settings to verify its applicability. Specifically, we study the performance of our proposed method with higher strengths, using a different covering array generation tool, and applying on real world benchmarks.

### *Higher strength*

We examine the behavior of our proposed method in higher strengths, 4 and 5. Since the generation time by ACTS becomes very long rapidly a long with degree and it takes more than 20 to 30 min for one execution, the experiment was limited in degree and constraint sets. In strength 4, the maximum degree was 60. In strength 5, the maximum degree was 40 and it was not possible to conduct the experiment with the constraint set ''BASIC+''.

Tables 8 and 9 show the generation time and the output size when $t = 4$ and $t = 5$ respectively. When the strength $t$ is 4, the generation time reduction (50-62.4%) was

**Table 8  Covering array generation performance; scratch; $t = 4$.**

| Constraint set | Degree | ACTS | | ACTS + proposed method | | Size penalty | Time reduction |
|---|---|---|---|---|---|---|---|
| | | Size | Time [msec] | Size | Time [msec] | | |
| NONE | 20 | 1,134 | 990 | 1,405 | 2,259 | 23.9% | 128.2% |
| | 40 | 2,027 | 196,226 | 4,649 | 73,864 | 129.4% | −62.4% |
| BASIC | 20 | 1,236 | 8,756 | 1,958 | 4,884 | 58.4% | −44.2% |
| | 40 | 2,041 | 247,151 | 4,261 | 123,471 | 108.8% | −50.0% |
| BASIC+ | 20 | 537 | 197,244 | 729 | 82,074 | 35.8% | −58.4% |
| | 40 | 945 | 909,183 | 1,817 | 453,700 | 92.3% | −50.1% |

**Table 9  Covering array generation performance; scratch; $t = 5$.**

| Constraint set | Degree | ACTS | | ACTS + proposed method | | Size penalty | Time reduction |
|---|---|---|---|---|---|---|---|
| | | Size | Time [msec] | Size | Time [msec] | | |
| NONE | 20 | 5,746 | 12,771 | 6,187 | 54,122 | 7.7% | −50.1 |
| | 40 | N/A | N/A | 45,108 | 4,773,071 | N/A | N/A |
| BASIC | 20 | 6,192 | 152,885 | 8,637 | 86,623 | 39.5 | −43.3 |
| | 40 | N/A | N/A | N/A | N/A | N/A | N/A |
| BASIC+ | 20 | N/A | N/A | N/A | N/A | N/A | N/A |
| | 40 | N/A | N/A | N/A | N/A | N/A | N/A |

observed at the cost of the size penalty (35.8–129.4%) excepting when the degree is set to 20 and no constraint was given. In strength 4, our method makes it possible to generate a covering array for degree = 40. ACTS was not able to generate an array directly for the degree. While the time reduction was still significant (43.3–50.1%), the size penalty became smaller (7.7–39.5%).

### PICT

As discussed in 'Flexibility of weaken product combinatorial join', the proposed method can employ any CIT tool for covering array generation. To make sure our method can be applied to other tools other than ACTS, we use PICT as the underlying covering array generation engine in this experiment and measures the performance of our approach with this different generation engine. In strength 2, PICT was not able to generate covering arrays when the defined constraint sets (*i.e.*, basic and basic+) were present even when the *degree* = 20 within 30 min. Also, when $t = 3$, it took more than 30 min to generate a covering array for degrees greater than 100. Therefore, our experiment with PICT only examine models without any constraint and we limit the degree for strength 3 up to 100.

Figures 24 and 25 show the results of comparing the generation time between our method with PICT and standalone PICT in strength 2 and 3 with no constraint) respectively. Tables 10 and 11 show the size of the generated covering arrays. We can see that the proposed method accelerates the covering array generation up to 76% and the size penalty was 16%–56% in strength 2. In strength 3, the acceleration was 96% and the size penalty

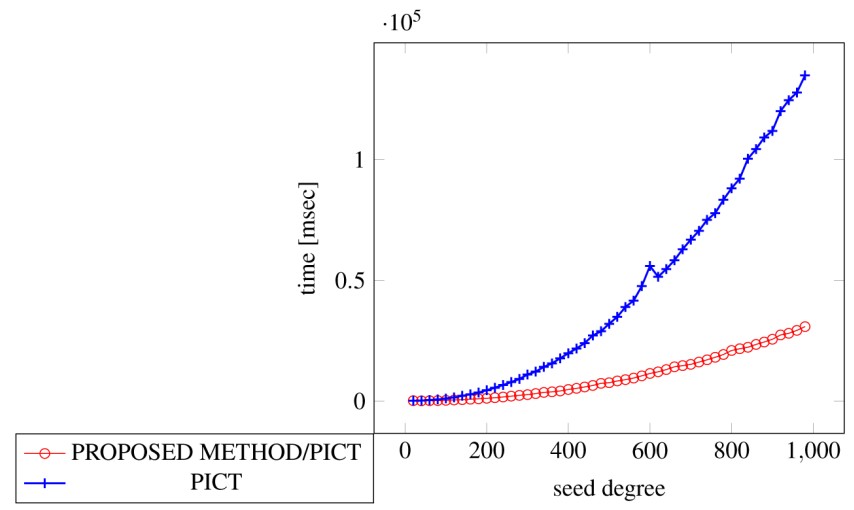

**Figure 24** Scratch generation; $t = 2$; constraint = none.

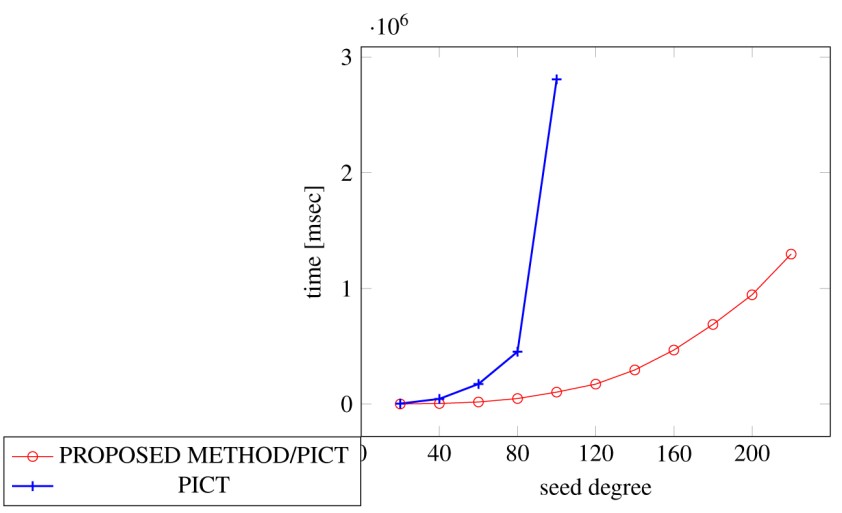

**Figure 25** Scratch generation; $t = 3$; constraint = none.

**Table 10** Size of covering arrays; scratch; $t = 2$; $d = [20, 980]$; *PICT*.

| Constraint set | None | | Basic | | Basic+ | |
|---|---|---|---|---|---|---|
| | min | max | min | max | min | max |
| PROPOSED METHOD based on PICT | 61 | 94 | N/A | N/A | N/A | N/A |
| PICT | 39 | 81 | N/A | N/A | N/A | N/A |
| Size Penalty with PICT | 56% | 16% | N/A | N/A | N/A | N/A |

was 71%. Given that the test cases are fully automated and they are executed overnight, so that the size penalty will not decrease the benefits of our approach in time reduction

**Table 11  Size of covering arrays; scratch; $t = 3$; $d = [20, 100]$; *PICT*.**

| Constraint set | None | | Basic | | Basic+ | |
|---|---|---|---|---|---|---|
| | min | max | min | max | min | max |
| PROPOSED METHOD based on PICT | 363 | 363 | N/A | N/A | N/A | N/A |
| PICT | 226 | 692 | N/A | N/A | N/A | N/A |
| Size Penalty with PICT | 60% | 71% | N/A | N/A | N/A | N/A |

**Table 12  Covering array generation performance; scratch; $t = 2$; *CASA*.**

| | ACTS | | ACTS + proposed method | | Size penalty | Time reduction |
|---|---|---|---|---|---|---|
| | Size | Time [msec] | Size | Time [msec] | | |
| APCHE | 33 | 939 | 60 | 712 | 45.0% | −31.9% |
| BUGZILLA | 19 | 499 | 28 | 476 | 32.1% | −4.8% |
| GCC | 23 | 719 | 30 | 698 | 23.3% | −3.0% |
| SPINS | 26 | 472 | 38 | 520 | 31.6% | 9.2% |
| SPINV | 45 | 644 | 84 | 630 | 46.4% | −2.2% |
| TCAS | 100 | 446 | 120 | 498 | 16.7% | 10.4% |

**Table 13  Covering array generation performance; scratch; $t = 3$; *CASA*.**

| | ACTS | | ACTS + proposed method | | Size penalty | Time reduction |
|---|---|---|---|---|---|---|
| | Size | Time [msec] | Size | Time [msec] | | |
| APCHE | 173 | 5151 | 269 | 3382 | 35.7% | −52.3% |
| BUGZILLA | 68 | 572 | 104 | 596 | 34.6% | 4.0% |
| GCC | 108 | 5,615 | 203 | 3,251 | 46.8% | −72.7% |
| SPINS | 98 | 497 | 186 | 516 | 47.3 | 3.7% |
| SPINV | 286 | 982 | 495 | 939 | 42.2% | −4.6% |
| TCAS | 405 | 488 | 471 | 537 | 14.0% | 9.1% |

### Real world benchmark

We also use a real world data model suite in our study, which is called CASA (*Cai, 2020*).
Table 12 shows the result of comparing the time to generate covering arrays and the sizes
of the generated covering arrays from the models contained in the real-world benchmark
data (*i.e.*, CASA) in strength = 2. As shown in the table, the generation time was reduced
up to 31.9% in strength 2. Table 13 shows the results of comparing the performance from
the same models in strength = 3.

We find that in strength 2, the largest acceleration is 31.9%, while the size increases
16.7–46.4% and the size penalty is in general larger in models whose degrees are smaller.
For example, the degree of SPINV is 55 and GCC degree is 199. Similarly, the method
accelerates the generation process up to 42%, while 16–90% increase in size is seen in
strength 3, and the penalty is larger in the models with smaller degrees.

## Summary and discussion

First of all, our proposed method offers a way to reuse test oracles designed in earlier testing phases (*e.g.*, unit testing, component testing) in later ones such as integration and system testing. Moreover, the method can accelerate covering array generation under complex constraint sets. It also enhances applicability of combinatorial interaction testing tools with different strengths and weaknesses in flexible combination, since the method is transparent to underlying generation algorithms.

Second, our proposed method accelerates an existing covering array generation algorithm by combining outputs of it, at the cost of increase in output size (*i.e.*, size penalty). In general, The size penalty becomes smaller and the time reduction becomes greater when the constraint set is more complex and the degree is larger. Specifically, the generation time reduction varies from 13% to 99% depending on generation scenarios and degrees of the method's output. The increase in size can go up to 141% in the worst case. Although the increase in size is big in some cases and it seems a concern to apply the approach, it is still beneficial, from different aspects in different situations, as discussed as follows.

A first simple practical use case is when the test execution time matters less than the test generation time, our proposed method will be useful, from a comprehensive perspective. This is common in modern software development projects, in which test execution is highly automated not only in unit testing but also in later testing phases such as integration testing and system testing.

Also, as shown in Figs. 6–11, the generation time of the conventional tool (*i.e.*, ACTS) grows more rapidly than linear, along with the degree increases, it becomes impractical quickly as the degree increases. Our approach first uses the tool to generate two smaller covering arrays and combines them later. This approach enhances the applicability of current generation tools (such as ACTS) to scenarios where it has not been practical due to too many parameters (*i.e.*, very high degree) and too long generation time. But with our approach, the large number of parameters are split into two sets, and the generation tool only handles half of the parameters, which may largely reduce the generation time and make it practical.

There are situations where the cost to change a value for a testing parameter varies largely. For example, some parameters require OS re-installation while others can be changed by just operating an application. If a single covering array (test suite) is generated for all the parameters at once, each test case may have different values of OS parameters, that may require an OS re-installation before executing each single test case, which can be very expensive. But our proposed approach can generate covering arrays for OS parameters and other application parameters separately and combine them into one later. This will largely reduce the switching of OS parameters (*i.e.*, OS re-stalling) to a minimal numbers. Since our method does not create a new row, the overall test execution cost will be reduced even with some size penalty, because the execution time of additional test cases is still much cheaper than the cost of many more times of OS re-installation.

When a user needs to add some parameters to an existing test suite, ''seeding'' functionality of a CIT tool has been used. As shown in Figs. 19 and 20, the generation

time is dramatically reduced when our method is applied to this use case. This is because the conventional method (ACTS) needs to examine coverage of the input covering array first, which is time consuming and unnecessary for our method. Since the size penalty for this use case is relatively modest (30%–60%), if test case design time matters more than execution time because of testing automation, for instance, it will be a practical solution.

Besides, more generally, in a situation where the constraints are more complex or the degree is larger, our proposed method shows more significant benefit, because the size penalty decreases and the time reduction increases as the constraint set becomes more complex and the degree increases.

The aforementioned situations show that our proposed method is beneficial, even with large size penalty at times.

## CONCLUSION

The "combinatorial join" operation, which was first introduced in *Ukai et al. (2019)*, combines two existing covering arrays to create a new covering array horizontally. In this paper, we proposed a novel algorithm called the "weaken-product based combinatorial join", which implements the operation.

We evaluated the algorithm from several aspects with regard to execution time and the size of an output array. We examined its performance in three scenarios as follows:

- Scratch generation
- Incremental generation
- VSCA generation

The improvements by our method in time efficiency were 33%–90%, 66%–99%, and 13%–34% respectively for Scratch, Incremental, and VSCA generation scenarios (RQ1). Although this method produces larger covering arrays than the conventional method, the increase in size remained reasonable in some practical use cases (RQ2). For instances, test execution is highly automated and the number of test cases less matters; the costs to change parameter values in test cases are very unbalanced; or several new parameters are added to an existing test suite.

In addition, our algorithm has other benefits as follows:

- Reusing test oracles across multiple testing phases (Oracle Reuse).
- Employing multiple covering array generation tools (Divide-and-Conquer).

For Oracle Reuse, we reviewed the discussion in the original paper (*Ukai et al., 2019*) and clarified the assumptions that were not explicitly stated. We identified three classes of bugs that can be detected by that approach, which are "resource conflict", "incorrect abstraction", and "unintended dependencies between components". To detect such bugs, a test suite for each component must be designed as independently as possible and must be fully described so that a consumer of the component can expect it to behave as defined by the reused oracle. The original paper asserted that the method can significantly reduce the total testing costs. We clarified that such a reduction is possible when the consumer-facing

component of the test suite can be reused as a system-level test for functionality testing (RQ3).

To evaluate the benefits of the "Divide-and-Conquer", we examined three well-known CIT tools, ACTS, PICT, and JCUnit. Specifically we evaluated their abilities to define test models, generation time and size efficiencies. Because existing tools have drastically different characteristics, it may be beneficial to apply multiple tools to construct one covering array. We had the following observations in this study:

1. Of the three CIT tools, ACTS was the fastest and produced the smallest covering array for factors without constraints or with simple constraints.
2. JCUnit had the most powerful notation to describe constraints for factors with a complex constraint set.
3. No single CIT tool is capable of handling software with industry scale and complexity.

As discussed in 'Flexibility of weaken product combinatorial join', testing parameters sometimes have quite different value changing costs. An OS-level parameter such as file system type might take hours to change, while an application level parameter value such as a text font type takes less than a second. In this situation, it becomes possible with this approach to generate an LHS covering array for OS parameters and RHS for application parameters and join them to construct a t-way-combination-covering test suite. This approach offers a way to guarantee t-way coverage among the OS parameters and application parameters without preparing a new OS installation nor executing all the test cases coming from the RHS(application) covering array on a configuration defined based on each row in LHS(OS).

Different CIT tools have different characteristics in terms of generation time, output size, and especially constraint describing capability. Our proposed method combines covering arrays regardless of the generation tools, therefore for each given input covering array (or sub-model), we may choose the most effective (*e.g.*, that can describe complex constraints) and efficient (*e.g.*, short time and small size) CIT tool for generation. In addition, different CIT tools may use different modeling languages to describe models, our proposed method does not require an universal modeling language to construct a single covering array, given its capability to combine all sub-models which may describe in different languages.

In summary, our approach can enhance the applicability of the CIT technique for software whose specifications are typically considered too complex for ACTS or too large for JCUnit (RQ4).

The proposed method delivers acceleration of covering array generation while it requires an increase in output size. It provides a new efficient option to generate a covering array for non simple use cases, such as incremental generations, input models with complex constraint sets, and VSCA generations. The increase in the size comes from the step to ensure all the input rows appear in the output (Step 3 in Fig. 1). We will improve this point to minimize the output size and the applicability of the method in our future works.

## Threats to validity

In this section, we discuss some threats to validity in our study.

### Internal validity

We designed the artificial model to simulate a situation where factors and constraints are automatically generated from a human friendly model. This model may not represent all practical situations, but in order to mitigate this, we tried combinations of different settings (*e.g.*, different testing strengths, degrees of parameters, and constraint sets, *etc.*) to cover as much as typical situations. We assumed that it is possible to convert such a high-level constraint into ACTS's notation in a short amount of time, which may not be true all the time.

### External validity

Our experiments were mainly conducted on synthetic data models. The intention was to simulate tools that generate a large number of factors and constraints from high-level models such as regular expressions and finite state machines. However, there is no general best practice for converting a high level model into an input parameter model and the data model we used might not reflect practical situations. To mitigate this, we conducted experiments using real world data sets called CASA.

The evaluation of the output sizes was based on the best practices and experiences of the first author's development team for an industry-scale software product. The conclusion may not be applicable to teams and/or other software products in different sizes.

### Conclusion validity

We did not conduct statistical verification over our experiments results and this can be a threat to conclusion validity. However, the elements involved in the experiments all consist of deterministic algorithms and we do not need such a procedure for the output sizes. On the other hand, the generation time grew monotonically along with the degree, except for only a few certain scenarios, such as scratch generation in $t = 3$ and degree is 340. Hence we do not consider it as a major threat in our conclusion.

## Future work

Our approach assumes that there is no constraint defined across *LHS* and *RHS*. However, it is usual not to have such an assumption in practical situations, especially when we construct a VSCA for a system with multiple components. From the technical point, sometimes it is even impossible to define a combinatorial join operation when constraints across *LHS* and *RHS* are present. For instance, if the strength of *LHS* and *RHS* is $t$ and there is a constraint across them which involves more than $t$ parameters in either *LHS* or *RHS*, there might not exist sufficient rows to cover all $t$-way tuples or even any row that satisfies the constraint at all. As one of our future works, we intend to study the exact criteria where the operation can be meaningful, and design an efficient algorithm to perform the operation under the situation that satisfies such criteria.

Our approach generates covering arrays of larger size than other tools, particularly when the strength is higher than 2. As one of the future work, we intend to apply a squashing technique to diminish a redundant covering array.

Lastly, our current algorithm is sufficiently fast in strength 2 and 3, but it may become less efficient in strength 4 or greater. It is known that a bug can be found in a strength up

to 6 or 7 (*Kuhn, Kacker & Lei, 2016*). Therefore, in order to improve the applicability of our approach in practice in high strength, our algorithm needs further improvement.

### Funding
The authors received no funding for this work.

### Competing Interests
Hiroshi Ukai is employed by Rakuten, Inc. The other authors declare that they have no competing interests.

### Author Contributions
- Hiroshi Ukai conceived and designed the experiments, performed the experiments, analyzed the data, performed the computation work, prepared figures and/or tables, authored or reviewed drafts of the paper, and approved the final draft.
- Xiao Qu, Hironori Washizaki and Yoshiaki Fukazawa analyzed the data, authored or reviewed drafts of the paper, and approved the final draft.

### Data Availability
   The tool to execute our experiments and the results and a description of the tool.

### Supplemental Information
Supplemental information for this article can be found online at http://dx.doi.org/10.7717/peerj-cs.720#supplemental-information.

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
