# Peer review of "Accelerating covering array generation by combinatorial join for industry scale software testing"

_PeerJ Computer Science, doi:10.7717/peerj-cs.720_

## Round 0.1 · original submission · Major Revisions

Please note that one of the reviewers suggests outright rejection and the other recommends a major revision. Both reviewers point out major issues in the methoology, which need redesigning the experiments and carrying out more structured and more extensive experiments. Hence, a major revision should not only involve the presentation issues, but more improtantly, it requires addressing fundamental methodological issues.

Reviewer 1 ·

Basic reporting

The paper is very difficult to understand due to many spelling and grammar mistakes:

Spelling mistakes on line 24 (compbinatorial-> combinatorial), line 144 (chat->that), line 617 (movers-> moves), and line 834 (Bbut->but).

The listings in the paper are not numbered and referred. A good way is to cross-reference the listings such as use ”Listing X shows something” instead of “in the listing bellow”.

On line 930, the actual percentages are missing and instead question marks are shown. It could be that the authors forgot to write the actual numbers.

Why is “weaken-product based combinatorial join” in quote marks?

“There are three streams of methods 47 have been studied” should be “There are three categories of methods that have been studied”.

Many of the figures are also too small to be readable (e.g., Figure 1, Figure 14, 15 etc.).

The obtained data is provided, but I expected a more thorough experimental setup to be provided, including how the data should be obtained from these tools, the input models used and so on. In this form, the results are not really replicable.

Experimental design

The experimental study relies only on some rather simplistic examples. Hence, whether the results can be generalized remains open to some extent. An extension to include more examples/higher strengths and input space models of larger size would be highly recommended. I am missing the results for PICT in Section 5. Causality and dependence between different variables is not thoroughly investigated.

Research questions should be rewritten since these contain mistakes and I am more confused after reading them. For example, the first research question actually contains two sub-questions. The first question is binary in nature, which seems quite simplistic given the limitations of such an investigation. No relation between these proxy measures of test efficiency and actual cost measures are given. Some useful advice can be found here: http://www.cse.chalmers.se/~feldt/advice/guide_to_creating_research_questions.pdf

The structure of the evaluation methodology and results sections could benefit if these are explicitly aligned with the structure proposed in the guidelines for conducting case study research in software engineering by Runeson et al. who considers for instance sections on the data collection procedure and the data analysis procedure (which seem to be implicitly mentioned in the paper).

The threats to validity section needs to be extended with regard to:

Conclusion validity. Does the treatment/change we introduced have a statistically significant effect on the outcome we measure?

Internal validity. Did the treatment/change we introduced cause the effect on the outcome? Can other factors also have had an effect?

Construct validity. Does the treatment correspond to the actual cause we are interested in? Does the outcome correspond to the effect we are interested in?

External validity. Transferability Is the cause and effect relationship we have shown valid in other situations? Can we generalize our results? Do the results apply in other contexts?

Credibility. Are we confident that the findings are true? Why? Dependability Are the findings consistent? Can they be repeated?

Confirmability. Are the findings shaped by the respondents and not by the researcher?

The relationship between the independent and dependent variables is not really explained in the context of the research questions. Would be good to explain the stated hypotheses on test efficiency and effectiveness. How about the time needed to generate the existing covering arrays that are used for the combinatorial join?

The experiment seems planned in a very ad-hoc manner. For example, the execution time measures are rather unscientific, and there is no possibility for any causality or actionable results. Potentially, the algorithm could be useful and the questions are interesting. However, IMHO it's unfortunately back to the drawing board for designing and reporting the experimental evaluation.

Validity of the findings

The generation of covering arrays is an interesting area of research in combinatorial testing. This paper is proposing an algorithm to accelerate the generation of covering arrays using joins. The algorithm takes two existing covering arrays of the same size and a strength t, to perform the join “operation”. First, the strength of the input covering arrays is reduced, and a cartesian product is taken between them. The final covering array with unused rows is then merged to obtain the output. The algorithm is then evaluated based on the time taken, size of covering array, constraints handling capability, and capability of variable strength covering array generation. The paper also discusses the potential oracle reuse, enabled by the proposed algorithm.

Covering array generation is an open area of research. Authors have done a good job of explaining how the approach can be used with a diverse set of CIT tools for the input covering arrays generation. The evaluation includes most generation scenarios. Generally, the paper covers well the background of the area. The authors explain quite well the approach, but there are major points that must be addressed.

While RQ1 shows promising results, the answers to the other research questions are not convincing. For example, RQ2 clearly shows that if t is 2, the size of the covering array generated by the proposed approach is at least 47% larger than the other approaches. If t is 3, the size increase is dramatic (around 264% at least). The authors’ argument in explaining this finding is that test execution is automated anyway. Even if the approach reduces the generation time by 11 to 88% (which would be a few minutes), this could add hours of test execution time.

Section 5.3 discusses how this approach can help in reusing test oracles for three kinds of faults. Concrete examples are needed here for each of the defects and for which oracle can be reused. Also, the answer to RQ3 is not justifiable unless some results are provided demonstrating that reuse of test oracle using combinatorial join has indeed caught new bugs.

The algorithm assumes that the covering arrays have the same number of columns, which makes the approach impossible for many realistic scenarios, e.g., regression testing scenarios or evolving software. The practicality of this algorithm needs to be discussed.

The creation of basic and basic+ baselines is questionable. For example, how did the authors end up with formulae 16 and 17 for defining the constraints? How are these constraints really representing the constraints in some real applications? The algorithm also does not handle inter-covering array constraints that might limit the applicability of the approach.

The running example for the demonstration of the algorithm considers two covering arrays with the same values and parameters. It makes sense to use a better example with different values in the covering array.

Reviewer 2 ·

Basic reporting

1) This paper seems to be an extension of the authors’ previous paper (Ukai et al. 2019), but the difference is unclear. The authors should better discuss and highlight the novelty of this study.

2) There is no background of covering array (e.g., definition, notation, example, etc.), as well as the constraints in combinatorial interaction testing. This paper should include them to make it self-contained.

3) This paper discusses the different modelling languages used by different tools to describe constraints. I do not think the proposed algorithm makes a contribution on this aspect.

4) Figures 1 and 2 are hard to read; the p_{i,k} notation in Equation (16) is unclear.

5) This paper is too long. For example, the experiment design is too verbose; Figures 3, 4, 5, and 6 are unnecessary.

Experimental design

1) The largest problem instance used in the experiment has only 200 parameters, which is far away from the “industry scale” that might have thousands of parameters.

2) There are many real-world benchmarks for covering array generation, but the experiment includes synthetic problem instances only.

3) The experiment investigates coverage strengths two and three. The reduction of covering array generation time is more important and beneficial for higher strengths, say five and six.

4) As discussed in related works, there are also algorithms that seek to reuse existing covering arrays to generate new arrays. The proposed algorithm is not compared with them in the experiment.

Validity of the findings

1) The proposed algorithm assumes that there is no constraint across pre-constructed covering arrays. This is a very strict assumption, so the applicability of the proposed algorithm in the real world is limited.

2) The results are not very promising. Although the proposed algorithm can reduce the time cost of covering array generation, it produces much larger arrays, and accordingly, much higher testing cost.

3) There is no data to support the claim of the answer to RQ3.

Additional comments

This paper addresses a very important topic in Combinatorial Interaction Testing (CIT). The proposed algorithm, which seeks to reuse existing covering arrays, could be an interesting direction in the field of CIT. However, this paper is too long; further improvements are also needed for the experiment design and discussion of results.

---

## Round 0.2 · Major Revisions

Both reviewers agree that the manuscript has undergone a major improvement and they appreciate your effort in doing so. However, as you see below, Reviewer 1 indicates major outstanding issues both in terms of presentation and also in terms of reproducibility of the results.
Please take these comments carefully into account when producing your second revision.

Reviewer 1 ·

Basic reporting

Even if the README file is useful, I could not find the suggested zip file, but only a latex project. I was unable to replicate the results. The README should also include details about the CASA benchmark and how one could use your approach on these.

Since the last review, several readability improvements have been made and some of the comments have been satisfactorily addressed. Would be good to emphasise in the summary of contributions the results related to RQ2 as you do for RQ1, RQ3, RQ4. Still, the listings in the paper are not numbered (e.g., Listing 1) and references in the text.

There are still places where I found writing issues. Another thorough proofreading would be helpful for fixing these issues (articles, punctuation). Figures 6 to 11 can be subfigures part of the same figure on scratch generation. The same for the other figures related to VSCA generation and incremental.

Just a few examples:

“superrior” should be “superior”

“ with a focus in their 186 different characteristics in defining constraints” should be “with a focus on their 186 different characteristics in defining constraints”.

“can be expressed using the < and negate operator” should be “can be expressed using the < and negate operators”.

“based combinatorial 595 join approach.” should be “based combinatorial 595 join approach:”

Experimental design

Improvements have been made to include input space models of larger size and the results for PICT. Substantial changes have been regarding the research questions. Substantial evaluation experiments have been made using the CASA benchmark. Unfortunately, not too many details about this benchmark are given. Also, the link in Shaowei Cai (2020) does not work. Shaowei Cai (2020). A description of this benchmark and some “size” metrics should be given in order to assess how realistic these benchmarks are.

In Section 4 one should give more details on the data sets used in terms of size, how relevant they are, variability and so on. A specific section on the selection of the data sets should be included.

Several improvements have been done in the threats to validity section.

Validity of the findings

Good clarifications are given and I appreciate the improvements made to the algorithm as well as the discussion related to a trade-off analysis between test execution time and the number of test cases. The examples of oracles classes are useful in the discussion of fault detection capability.

Would be good to clarify how the two approaches outlined in 4.3.1 Generation Scenario are useful in practice. When would an engineer using your approach benefit from generating a CA from scratch incrementally? Clarify the statement “Our tool makes reuse of oracles with higher possibilities”.

Section 5 could benefit from better explanations and interpretations of the results shown in each Table and Figure.

Additional comments

I notice with satisfaction that most of the comments made in my first review have been taken into account. Nevertheless, there are still some issues that remain to be resolved.

Reviewer 2 ·

Basic reporting

The presentation of the revised version is now clear and self-contained.

Experimental design

The experiment design is improved. Especially, real-world problem instances are now included.

Validity of the findings

The potential threats to validity have been well discussed.

Additional comments

The authors have paid good efforts to address my previous comments, and I think this paper is now ready for publication.

---

## Round 0.3 · accepted · Accept

It appears from the review report that the comments on the previous revision have now been satisfactorily addressed and
hence, I am happy to accept the paper.

Reviewer 1 ·

Basic reporting

Since the last review, several improvements have been made.

Experimental design

I notice with satisfaction that most of the comments related to the experiment design have been taken into account.

Validity of the findings

The paper now appears as a well-rounded piece of work.